# A self-regulating shuttle for autonomous seek and destroy of microplastics from wastewater

Dennis Kollofrath, Florian Kuhlmann, Sebastian Requardt, Yaşar Krysiak ⓘ & Sebastian Polarz ⓘ ✉

Microplastics pose a significant environmental challenge, causing harm to organisms through inflammation and oxidative stress. Although traditional adsorbents effectively capture pollutants, they are limited by their localized action and require laborious recycling processes. We introduce a buoyancy-driven hybrid hydrogel that functions as a self-regulating shuttle, capable of transporting and decomposing contaminants without external intervention. By leveraging thermally switchable buoyancy, the material cyclically ascends from the seabed to the water surface, facilitating pollutant degradation, before descending to restart the process. This motion is enabled by vinyl-functionalized porous organosilica and thermoresponsive poly(N-iso-propylacrylamide) (pNIPAM), which allow for reversible gas bubble storage and precise control over ascent and descent dynamics. As a demonstration, we apply this platform to microplastic decomposition, where light-induced reactive oxygen species effectively degrade collected particles. Adjustments to catalyst concentration further optimize transport kinetics, enhancing efficiency across various conditions. While microplastic remediation showcases its capabilities, this shuttle represents a broadly adaptable system for sustainable pollutant removal and environmental remediation.

The increasing development of modern societies leads to a growing accumulation of discarded products and other waste materials. With expanding populations and economies, the long-term trajectory of environmental pollution remains uncertain. Addressing this challenge requires continuous innovation in material design to develop efficient and adaptable strategies for contaminant removal. Depending on the type of waste and its dispersion within a medium, purification processes can be complex and inefficient—microplastics are a prime example. The high stability and low biodegradability of polymeric materials, once considered key advantages, have now become long-term environmental concerns. Over time, mechanical and chemical degradation processes generate micro- and nanoscale plastic particles that persist in ecosystems and enter the human body through ingestion and respiration[1,2]. Aquatic systems are particularly affected, with microplastics detected at all depths, from surface waters to deep-sea sediments[3–8].

Most microplastics consist of polyethylene (PE), polypropylene (PP), polyvinyl chloride (PVC), and polystyrene (PS), and their negative effects on living organisms are well-documented[9–14]. Beyond their direct impact, microplastics can serve as carriers for toxic pollutants such as heavy metal ions, antibiotics, and pesticides, further amplifying their environmental risk[15–18]. Notably, aged microplastics have been found to be more readily absorbed into cells than newly introduced particles[19]. Given the scale of this issue, various approaches are currently being tested for the purification of wastewater from microplastics. Traditional separation techniques, such as membranes[20], agglomeration[21], electrocoagulation[22], and activated sludge[23], have notable drawbacks, including high energy consumption, limited recyclability, and an inability to effectively capture small pollutants like nanoplastics[24]. Furthermore, their applicability in open-water environments is restricted by

Institute of Inorganic Chemistry, Leibniz Universität Hannover, Hannover, Germany. ✉e-mail: sebastian.polarz@aca.uni-hannover.de

instrumental complexity. This highlights the need for novel, adaptable approaches.

Recent advancements have introduced micro-/nanobots for microplastic degradation, utilizing reactive oxygen species (ROS) for chemical breakdown[25–27]. For a comprehensive overview, the reader is referred to the review articles by Pumera et al.[28] and Parmar et al.[29]. The described systems are externally controlled, requiring the initial identification of microplastics and their location. Once this is determined, microswimmers can be directed to the target area, with various fuels introduced to drive the chemical processes. This is only possible if the destination is also reachable from the outside. However, depending on their size, (polystyrene) microplastics can accumulate near the sediment due to their density of up to 1.1 g/cm³[30]. Due to the anoxic milieu and darkness in deeper waters, it becomes challenging for the microswimmer to reach the bottom, making the oxidative decomposition of the microplastics impossible[31]. An autonomous system that transports contaminants from the seabed to the surface, where they are decomposed, would represent a breakthrough in reprocessing wastewater.

Here, we present a buoyancy-driven hybrid hydrogel designed as a generalizable platform for contaminant transport and remediation. To illustrate its potential, we investigate its application in microplastic decomposition. As outlined in Fig. 1, the material autonomously cycles between the seabed and the water surface, collecting microplastics at depth and transporting them to the surface, where ROS-induced degradation occurs. This fully autonomous, fuel-driven vertical motion − enabled by the reversible capture and release of gas within the nanoporous organosilica network − is governed by the interplay of a temperature-responsive hydrogel scaffold, fuel-reactive porous organosilica, and an embedded photosensitizer. Following each cycle, the hydrogel re-descends, allowing for continuous operation. This study demonstrates how a single material can integrate multiple functionalities−buoyancy modulation, selective capture, and controlled degradation−into a self-regulating purification system. While we explore its use in microplastic removal, the underlying design principles could be adapted for other environmental remediation challenges.

## Results

### Synthesis of buoyancy-driven shuttle (BDS) gels and microplastic uptake

In previous publications by our group, we have prepared nanohybrids composed of vinyl-functionalized, porous organosilica particles and functional polymers, such as thermoresponsive systems[32,33]. Furthermore, we have gained extensive experience in synthesizing porous materials for ROS generation[34,35]. This knowledge, along with other recent literature inventions, is implemented in the material presented here (see Fig. 1 - bottom part)[36,37]. The synthesis of the materials is described in detail in the experimental section, and additional analytical data are provided in the Supplementary Information (Supplementary Fig. S1). The material's scaffold is a macroporous pNIPAM (poly(N-isopropylacrylamide)) gel, in which the vinyl-functionalized organosilica nanoparticles act as crosslinkers. The vinyl groups are also used to introduce the drive unit and the photosensitizer to the particles (Fig. 2).

To ensure efficient photothermal heating for the sunlight-driven phase transition of pNIPAM, the BDS-gel is further coated with a black polydopamine layer (Fig. 3a). This coating not only protects the polymer chains from self-produced reactive oxygen species[38–40] but also enhances the gel's ability to bind microplastics across a broad pH range, thanks to the strong adhesive properties of polydopamine[41]. By reducing the zeta potential of the hydrogel surface, the polydopamine layer significantly improves the uptake of polystyrene particles[42–44]. Although both the polystyrene beads (Supplementary Fig. S2a) and the hydrogel surface are expected to have negative zeta potentials[45–47]

adsorption is facilitated by a combination of non-electrostatic interactions, including hydrophobic and π−π interactions. Additionally, the presence of salt in solution (seawater) screens electrostatic repulsion, further promoting the adsorption process. The resulting dark hybrid hydrogel and its thermo-responsive properties are shown in Fig. 3b. A scanning electron microscope measurement of the material shows the porous structure of the gel and the cross-linked NOPs embedded on the pore walls (Fig. 3c, d). Below its lower critical solution temperature (LCST) of 32 °C, the hydrogel absorbs water from its environment and swells, a behavior that persists even at very low temperatures down to 4 °C[48,49]. If the temperature rises above the LCST of pNIPAM, the gel begins to collapse, losing about 90% of its volume. The extreme volume change during swelling creates suction in the gel, greatly increasing diffusion and thus enabling the adsorption of microplastics.

The microplastic decomposition cycle (Fig. 1, Phase 1) begins with the uptake of microplastics into the BDS gel. Guo et al.[50] showed that pNIPAM hydrogel actuators can serve as adsorbents for microplastic pollutants. We have selected polystyrene beads (diameter 2 μm) as a model contaminant since it is one of the most widely used types of plastic and poses significant potential human health risks from food chain exposure routes influenced by marine waters[51–53]. To demonstrate the BDS gel's capacity to adsorb microplastics, it was immersed in an aqueous polystyrene dispersion prepared with a 3.5% sodium chloride solution to simulate seawater. Initially, the gel was in its collapsed state (Fig. 3b, left) and was allowed to swell.

The uptake of polystyrene microbeads can be visualized by SEM micrographs (Fig. 3e), as well as infrared spectroscopy (IR) (Fig. 3f). The vibrational bands at 696 cm⁻¹, 1494 cm⁻¹, and 3026 cm⁻¹ can be assigned to the aromatic ring and C-H vibrations of polystyrene. Quantitative evaluation by thermogravimetric analysis (TGA) (see Supplementary Fig. S2b) resulted in a microplastic intake of 57 mg by a gram of hybrid hydrogel. This achievement alone is remarkable; however, it is important to note that this capacity can be utilized repeatedly in each cycle. TGA and infrared spectroscopy measurements before and after multiple washing cycles−including heating, collapsing, and swelling−confirm that the microplastic remains firmly adsorbed to the hydrogel with no detectable leakage (Supplementary Fig. S2b, c). This is particularly important, as the system must prevent the unintended release of microplastics at the water surface due to elevated temperatures during operation.

### Implementation of the buoyancy drive into the NOPs and ascending process

In the subsequent phase of the cycle (Fig. 1, Phase 2), the gel rises to the water surface. This buoyancy-driven movement is aided by the pore system of the NOPs, which generates and reversibly accumulates gas bubbles from D-glucose. This system utilizes a tandem of two catalysts: glucose oxidase (GOx) and platinum nanoparticles (Pt-NPs). Glucose oxidase catalyzes the conversion of D-glucose into D-gluconic acid and hydrogen peroxide across a broad pH range, which is essential for the buoyancy mechanism[54]. Meanwhile, Pt-NPs convert the hydrogen peroxide into oxygen bubbles and water (Fig. 4a, b).

To integrate this buoyancy mechanism, Pt-NPs with an average diameter of 1.9 nm were synthesized according to established protocols[55] and incorporated into the NOPs' pore system (Supplementary Fig. S3a, b). Previous studies demonstrated that wet impregnation effectively functionalizes pores with various organic molecules[32,33], and we have now extended this technique to metal nanoparticles. Subsequently, GOx was immobilized via a UV-catalyzed thiol-ene click reaction between the vinyl groups on the NOPs and the cysteine moieties of the enzyme, resulting in Pt/GOx-NOPs (Fig. 4, Supplementary Figs. S3, S4). Characterization using TEM tomography (Fig. 4c, Supplementary Fig. S3c) confirmed the even distribution of Pt-NPs throughout the NOPs. N₂-physisorption analysis showed that Pt-NPs were localized within the pore system, evidenced by a notable

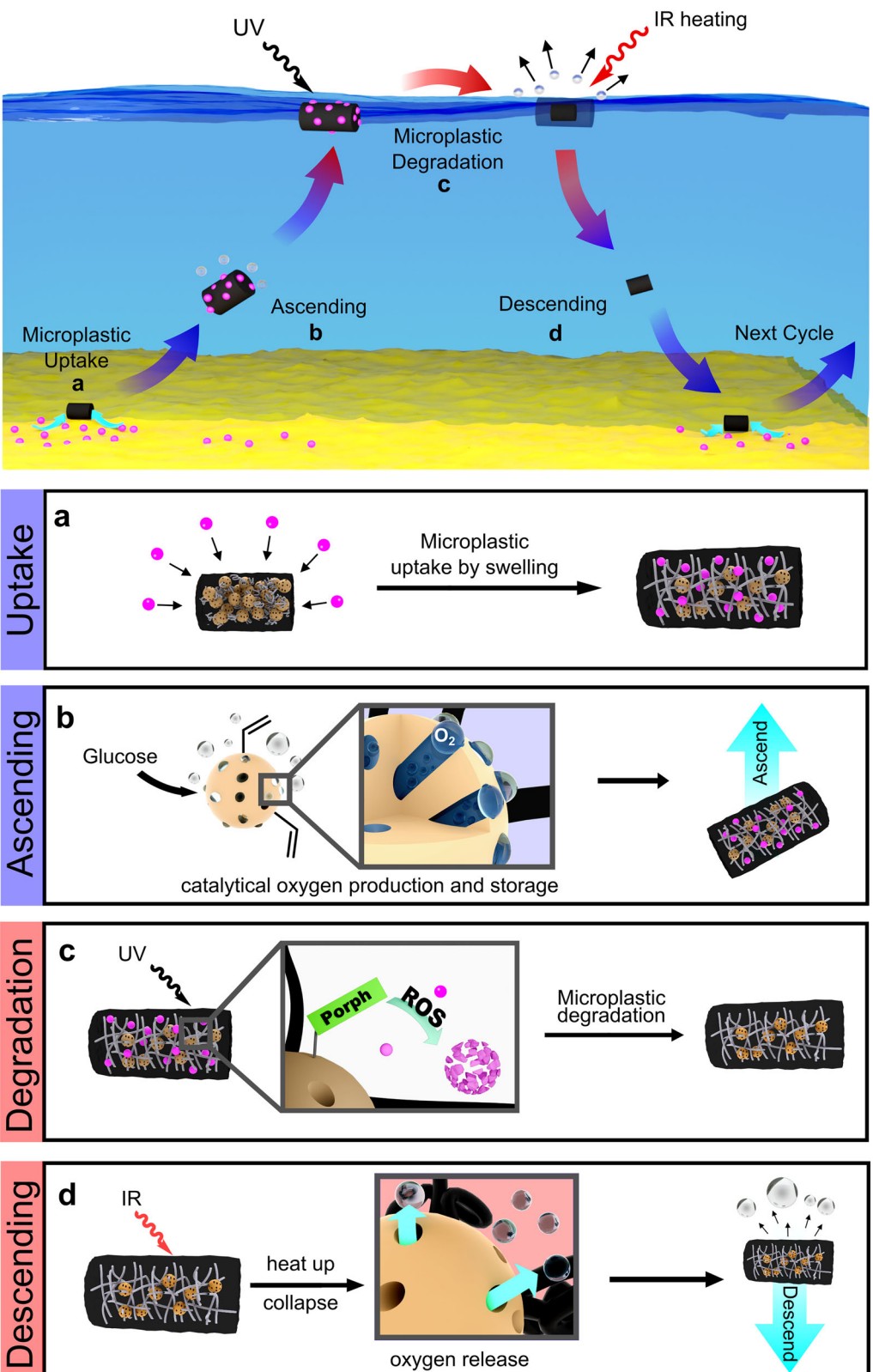

**Fig. 1 | Concept of an autonomous buoyancy-driven shuttle (BDS) gel for the degradation of microplastics.** It consists of thermoresponsive pNIPAM cross-linked with nanoporous organosilica particles (NOPs). **a** At the seabed (cold), the collapsed hybrid hydrogel begins to swell and adsorb microplastics. Simultaneously, due to increased accessibility during swelling, the drive unit produces oxygen from D-glucose. **b** The oxygen formed is captured in the pores of the NOPs. Once a critical amount of oxygen has been generated, the hydrogel begins to ascend due to buoyancy caused by the oxygen. **c** Two light-induced effects occur at the water's surface. Firstly, the photocatalyst (porphyrin) on the NOPs generates ROS, which decomposes the collected microplastics (center), and secondly, the gel heats up due to its black color. **d** The introduced heat energy causes the BDS gel to collapse and the accumulated oxygen to escape. The gel descends, and the next cycle starts.

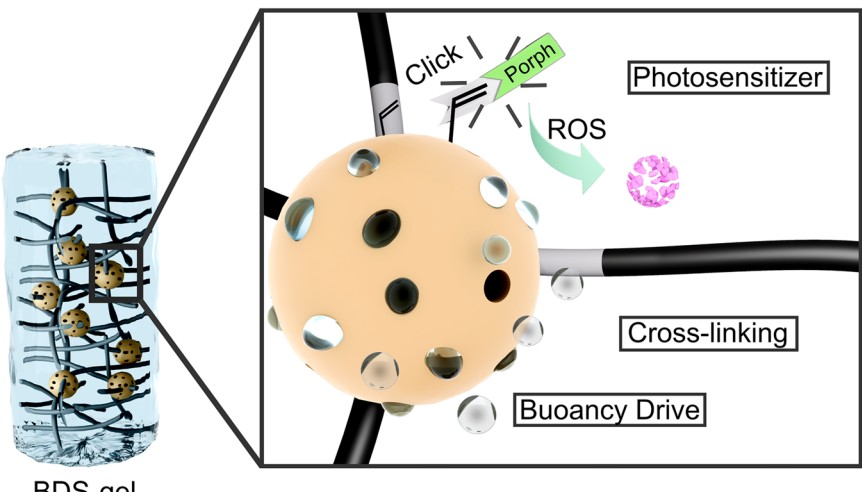

**Fig. 2 | Schematic representation of the buoyancy-driven shuttle gel made up of nanoporous organosilica particles (NOPs) and pNIPAM.** The NOPs serve multiple functions: they act as covalent cross-linking points within the hydrogel network, transport the porphyrin-based photosensitizer responsible for reactive oxygen species (ROS) generation, and incorporate the propulsion unit that enables the autonomous ascent and descent of the gel.

decline in porosity, particularly in pores larger than 4 nm (Supplementary Fig. S3c, d). This is reasonable, as the 1.9 nm Pt-NPs are more likely to diffuse into the larger pores of the NOPs. Thermogravimetric analysis (Supplementary Fig. S3f) indicated 0.9 wt % Pt-NPs in the NOPs, with no significant leaching after three washing cycles (Supplementary Fig. S3g).

An enzyme activity assay confirmed that GOx remained active after attachment, maintaining functionality between 10 °C and 40 °C, with peak activity at 40 °C (15 U/g) (Fig. 4d), consistent with literature values for the free enzyme[56]. To assess enzyme activity within the hybrid hydrogel, the assay was repeated with functionalized NOPs embedded in the gel, confirming that GOx remained active with an activity of 5.4 U/g (Supplementary Fig. S4f). The reduced activity in the hydrogel can be attributed to restricted diffusion within the gel matrix compared to the free NOPs. The successful D-glucose-to-oxygen reaction cascade was first verified using dispersed functionalized NOPs in a 1% D-glucose solution, where rapid oxygen bubble formation demonstrated their catalytic efficiency (Supplementary Fig. S4e). Finally, to validate the macroscopic functionality of the buoyancy drive, a hybrid hydrogel incorporating functionalized NOPs was immersed in a 1% D-glucose solution, confirming that the generated gas effectively induced flotation. The shuttle's ascent was recorded (Supplementary Movie S2), with key moments shown in Fig. 4e. Initially, the collapsed BDS gel descends to the vial's bottom. Upon temperature-induced swelling, glucose is absorbed, triggering the cascade reaction. Oxygen accumulates in the pore volume, reducing the gel's density below that of water, causing it to ascend.

By adjusting the catalyst content on the particles, the ascent duration of the shuttle gels can be optimized. Three batches of NOPs with varying platinum concentrations (0.2 wt%, 0.8 wt%, and 1.6 wt%) were used to create hybrid hydrogels (Supplementary Fig. S5a, b). When immersed in a 0.3% hydrogen peroxide solution, ascent times were recorded (Fig. 4f; Supplementary Fig. S5c; Supplementary Movie S1), showing that higher catalyst concentrations shorten ascent times. Specifically, a fourfold increase in catalyst concentration decreases the ascent time by a factor of four. Additionally, higher catalyst concentrations improve ascent speed by generating larger volumes of oxygen. Once the critical amount of oxygen required for ascent is attained, any excess oxygen can accumulate in the pore system of the particles, and occasionally in the polymer chains of the pNIPAM, further increasing buoyancy and accelerating the ascent.

## Microplastic degradation performance of buoyancy-driven shuttle gels

Upon reaching the water surface, the third step in the self-regulated microplastic decomposition cycle begins: the degradation of microplastics (Fig. 1, Phase 3). We selected photodegradation as the method for microplastic decomposition because it allows the system to function autonomously without needing external control. This process is driven by reactive oxygen species (ROS)—such as hydroxyl radicals (·OH), singlet oxygen ($^1O_2$), hydrogen peroxide ($H_2O_2$), and superoxide anion $O_2^{·-}$ - all of which are known to degrade microplastics, even down to molecular $CO_2$ effectively[25,57–60]. While the exact mechanism remains under investigation, the primary reaction pathways involve the addition of ROS, C-C-scissions, and H-abstractions[25,61]. Notably, Dutta et al.[62] used polyoxometalate-infused hydrogels to regenerate the gel by UV-light irradiation. In our system, we immobilize a photosensitizer on the surface of the NOPs, allowing sunlight irradiation to generate ROS, thus enabling the autonomous decomposition of microplastics. We chose the known singlet oxygen producer tetrakis-(4-carboxyphenyl)porphyrin (TCPP) as a photosensitizer as it can be easily modified with a thiol group to enable thiol-ene click chemistry for the functionalization of the NOPs (Fig. 5a; Supplementary Fig. S6a). Additionally, porphyrins have proven effective in decomposing polystyrene[63–68] and are known to have low cytotoxicity, making them ideal candidates for environmental applications[69–71]. After successful functionalization (Supplementary Fig. S6b), the ROS-producing properties of the Pt-NP/GOx/Porph@NOPs were assessed by the ROS-induced degradation of 9,10-anthracenediyl-bis(methylene)dimalonic acid (ABDA). ABDA rapidly converts to a steady-state endoperoxide product by reacting with singlet oxygen, and the decrease in its absorption band is observable by UV-Vis spectroscopy (Supplementary Fig. S6c, d), confirming that the functionalized NOPs are capable of generating ROS ($^1O_2$).

To demonstrate ROS-induced microplastic decomposition, the polystyrene-loaded BDS-gel was irradiated using a sunlight simulator (Fig. 5b). As a control, the experiment was also performed with a BDS-gel without the photosensitizer on the NOPs. This ensured that any observed microplastic degradation was due to ROS rather than photothermal heating.

After microplastic uptake, a portion of the gels was removed as a reference, while the remaining gels were irradiated. The samples were then lyophilized and analyzed using Raman spectroscopy, scanning electron microscopy (SEM), and thermogravimetric analysis. SEM micrographs (Fig. 5c–e) indicate that ROS treatment was successful, as

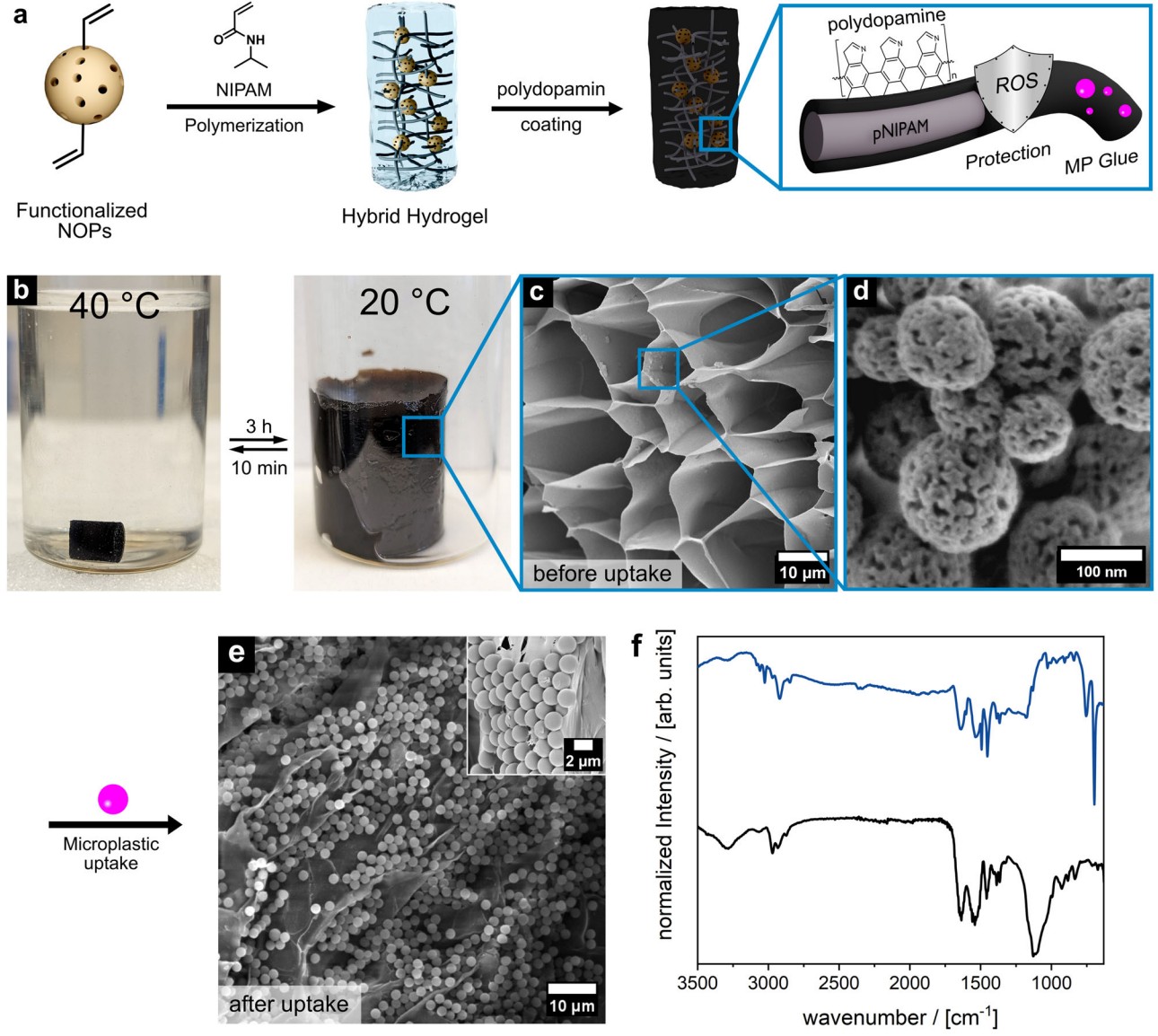

**Fig. 3 | Preparation and characterization of BDS gels and their microplastic uptake behavior. a** Overview of the preparation procedure for buoyancy-driven shuttle gels of NOPs and pNIPAM, followed by the coating with polydopamine. **b** Photographs of the BDS gel in the collapsed (left) and swollen (right) states. **c** SEM micrograph (scale bar = 10 μm) of the hybrid hydrogel composed of porous organosilica nanoparticles and thermoresponsive pNIPAM. **d** SEM micrograph showing the NOPs embedded in the pore wall of the hydrogel (scale bar = 200 nm). **e** SEM micrograph of the BDS gel after microplastic uptake (scale bar = 20 μm; inset: scale bar = 2 μm). **f** Infrared spectra of the BDS gel before (black) and after (blue) microplastic uptake.

very few microplastics remained on the sample containing the photosensitizer. In contrast, for the BDS-gel without the photosensitizer, no noticeable difference was observed before and after irradiation (Supplementary Fig. S7a). This is confirmed by thermogravimetric analysis of the gels before and after irradiation (Fig. 5f, Supplementary Fig. S7c). Results showed minimal microplastic degradation (4.1 wt%) in the control, compared to 98.7 wt% degradation with porphyrin-functionalized NOPs. This is additionally validated by Raman spectroscopy. Supplementary Fig. S7b shows the spectra before and after polystyrene uptake and irradiation of the BDS-gel. The discernible presence of vibrational bands at 1003 cm$^{-1}$ (symbolizing the ring breathing mode of the benzene ring) and 3058 cm$^{-1}$ (indicating the aromatic C-H vibration) following microplastic uptake is attributed to polystyrene. After irradiation, these distinct bands vanish completely, indicating that polystyrene is decomposed by the ROS.

To ensure that no smaller microplastic fragments or nanoplastics were released from the gel, we performed dynamic light scattering

(DLS) experiments on the washing solution after irradiation (Supplementary Fig. S9e). While DLS may not be the most optimal method for directly observing the degradation process[72], it is an effective technique for detecting particle fragmentation. The DLS measurements revealed no detectable smaller particles in the washing solution, indicating that no microplastics, whether original or fragmented, were released from the gel. This result, combined with the washing experiments with polystyrene- loaded BDS- gels (Supplementary Fig. S2b), further supports the conclusion that microplastics are degraded within the gel. These experiments demonstrated that no microplastics were released from the gel, even under conditions of heating, swelling, or collapsing, suggesting that thermally triggered desorption of the MP can be excluded. Moreover, the comparison with the control (without photosensitizer) confirms that the reduction in polystyrene content after irradiation is due to the reactive oxygen species (ROS) generated by the photosensitizer. While we cannot conclusively prove complete degradation to $CO_2$, these results demonstrate that the

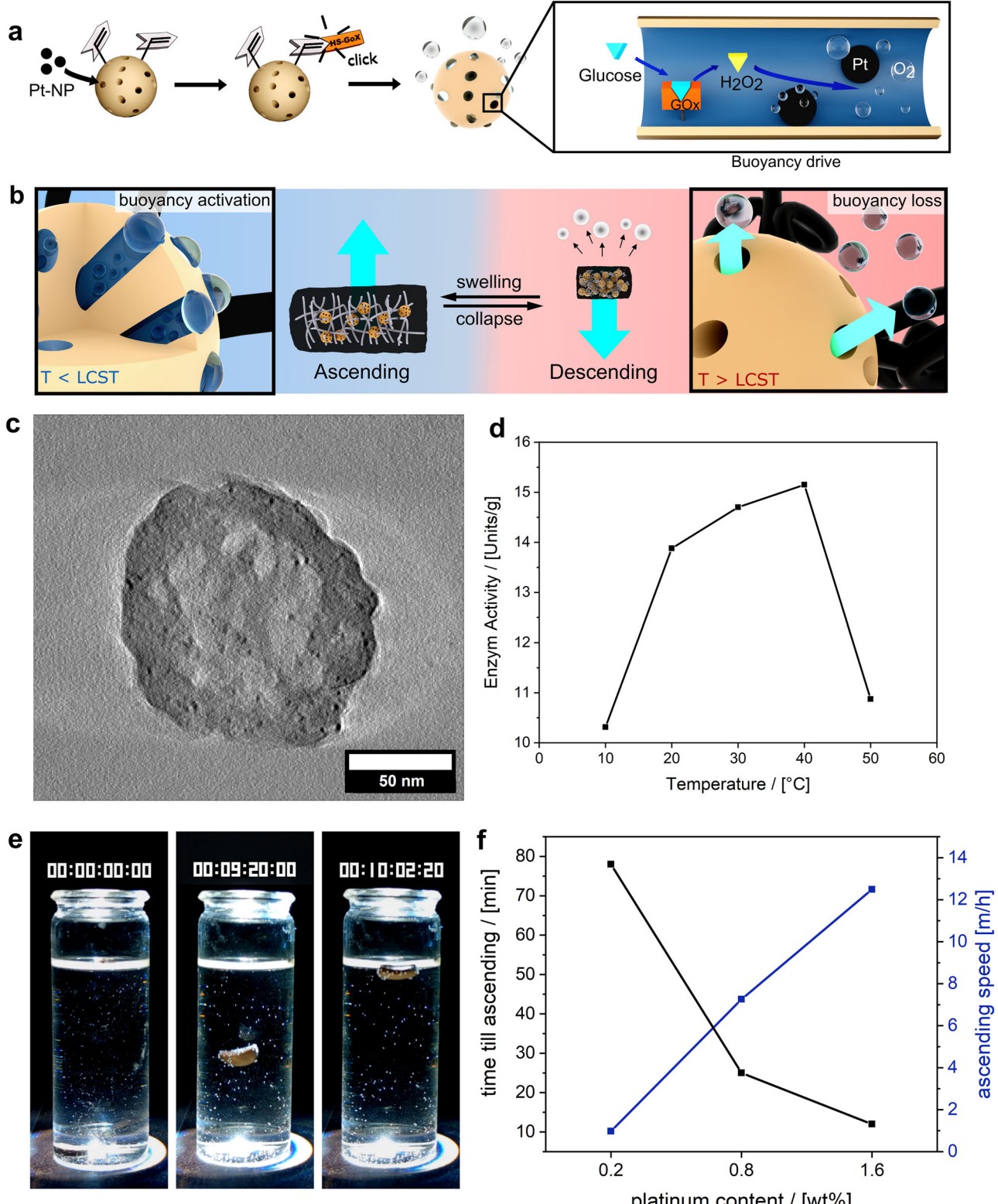

**Fig. 4 | Characterization of the buoyancy drive unit in the BDS-gel. a** Overview of the preparation procedure for the buoyancy drive unit of the BDS-gels. **b** Mode of operation of the buoyancy drive during the ascending process (left) and the descending process (right). When T < LCST of pNIPAM, the BDS-gel swells and absorbs glucose into its pores. The drive unit within the pores then converts the glucose into oxygen, which is stored in the NOPs. When T > LCST of pNIPAM, the BDS-gel collapses, and the polymer chains become more hydrophobic. This, combined with the expansion of the gas due to increased temperatures, facilitates the expulsion of the gas bubbles, thereby deactivating the buoyancy. **c** TEM micrograph of the Pt-NP/GOx functionalized NOPs (scale bar = 50 nm). This is part of a tomography shown in Supplementary Fig. S3c. **d** UV-Vis study of the enzymatic activity of GOx functionalized NOPs at different temperatures. **e** Photographs of the ascending process of the shuttle gel at different time intervals (hh:mm:ss:ms). **f** Comparison of the ascent time and speed of various shuttle gels in relation to the platinum concentration (determined from Supplementary Movie S1).

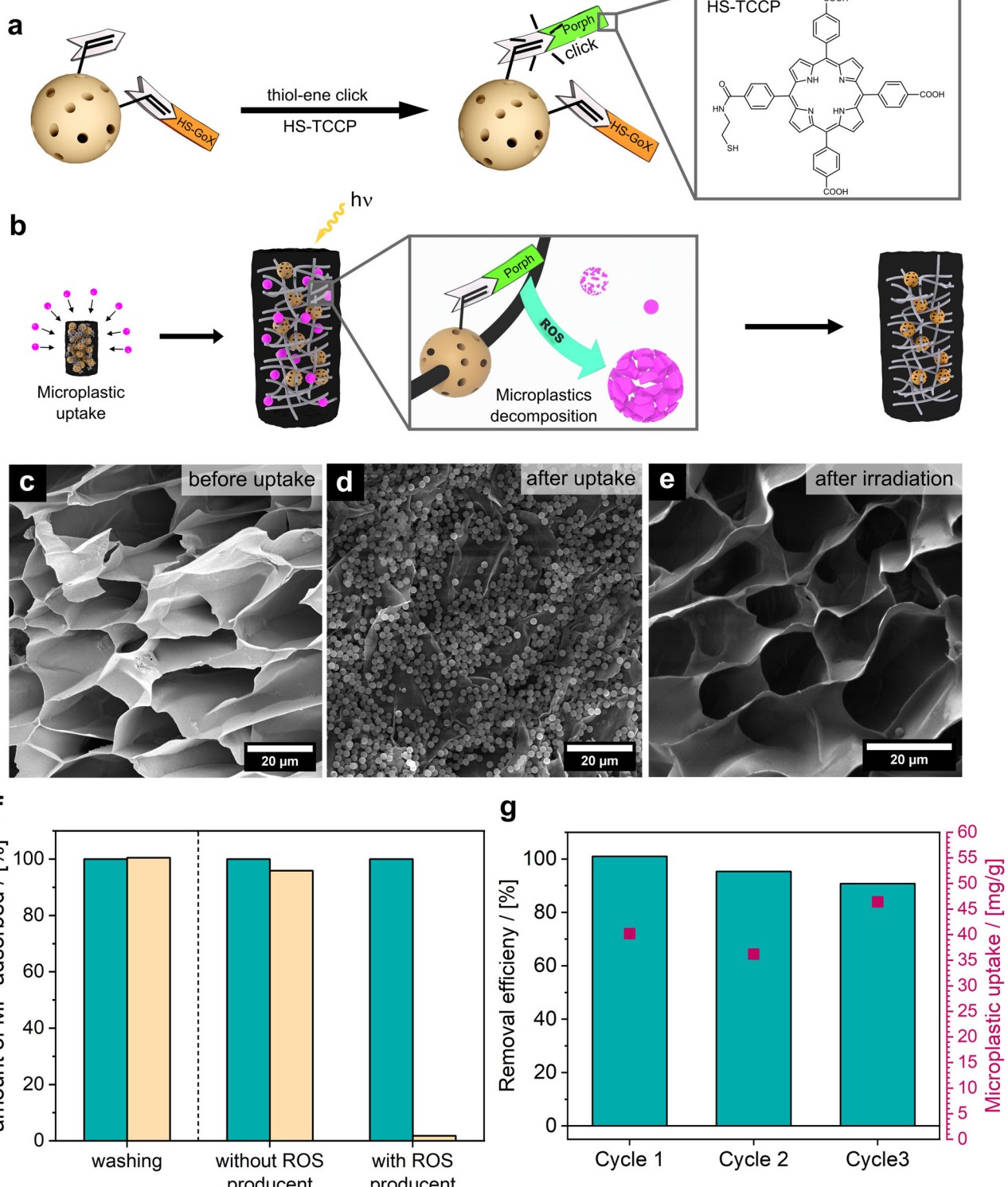

Fig. 5 | **Functionalization of NOPs with a photosensitizer, microplastic uptake and degradation performance of the BDS gel. a** Experimental procedure to functionalize the NOPs with the photocatalyst **b** Schematic illustration of the polystyrene uptake and its ROS-based decomposition. SEM micrographs before (**c**) and after (**d**) the microplastic uptake and after the irradiation. **e** (scale bars = 20 μm). **f** Thermogravimetric comparison of the microplastic quantity in the BDS-gel before (cyan) and after (yellow) washing (left) and before (cyan) and after (yellow) irradiation (right). A BDS-gel without porphyrin on the NOPs was used as a reference. **g** Microplastic removal efficiency (cyan) and total microplastic uptake (magenta) for three cycles determined from thermogravimetric analysis.

microplastic is no longer present within the gel after irradiation, and no degradation byproducts are found in the solution. This suggests that the microplastic has undergone at least partial decomposition into aromatic oxygenates within the 2 h irradiation time[67,73,74]. This is relatively fast compared to what is reported in the current

literature[68,75–77]. Our hypothesis is that this effect could be due to an additional advantage of our material—the porosity. Most systems known from the literature are non-porous (e. g., microswimmers, TiO₂ particles). The porosity at various length scales (nm range of NOPs, μm range of the hydrogel) might enable microplastics to remain in the gel

and at the active centers for a longer duration. Upon desorption, the microplastics are less likely to be released back into the water; instead, they could adsorb at a different location within the gel. Additionally, the degradation byproducts may remain trapped within the hydrogel matrix for extended periods. If this is the case, the efficiency of photodegradation might be enhanced by the combined effects of the porosity of the BDS hydrogel and the retention of both microplastics and their degradation byproducts[78–81].

For a sustainable application over several cycles, the BDS-gel itself mustn't be degraded by ROS. Infrared spectroscopy confirmed that ROS exposure does not harm the BDS-gel, as no new vibrational bonds appeared after the treatment (Supplementary Fig. S9a, b). Additionally, no optical differences were observed in the SEM images before and after irradiation (Fig. 5c–e). EDX measurements showed that platinum, silicon, and sulfur (from porphyrin and GOx) were still present (Supplementary Fig. S9c, d), and no changes were observed in the magic angle spinning (MAS) solid-state NMR spectra (Supplementary Fig. S9f). This ensures the BDS-gel maintains its structural integrity, allowing for repeated use in the microplastic degradation process. To test the reusability of the BDS gel, we loaded the gel with polystyrene three times and subsequently irradiated it. Gel samples were taken before loading, after loading, and after irradiation and analyzed using thermogravimetric analysis (Fig. 5g; Supplementary Fig. S8). Simultaneously, samples of the microplastic solution were collected, and the weight difference between cycles was determined by drying the solution to assess how much microplastic was removed from the solution. The results demonstrate that the gel can be used over multiple cycles without significant loss in removal efficiency. Furthermore, the amounts of microplastics removed from the solution closely match the quantity of microplastics remaining in the gel, indicating that the microplastic degradation is both efficient and reproducible.

### Autonomous descending of the buoyancy-driven shuttle gels

The final step in the cycle involves the autonomous descent of the BDS gel (Fig. 1, Phase 4). In addition to reactive oxygen species (ROS) production, sunlight irradiation induces photothermal heating of the gel above its lower critical solution temperature due to its black color (Fig. 6a). This heating causes the BDS gel to collapse (Fig. 6b–e) and results in the subsequent release of the oxygen stored in the pore system of the NOPs. If the amount of gas in the pore system drops below a critical level, the buoyancy is no longer sufficient to keep the shuttle at the water's surface, causing the gel to descend back to the cooler bottom of the vessel. There, the gel begins to swell again, and the cycle repeats. Three mutually favorable factors might contribute to the release of oxygen during heating. First, the heating causes the gas in the pores to expand, forcing it out of the pore system (Fig. 6f). Second, the previously hydrophilic poly(N-isopropylacrylamide) (pNIPAM) chains become more hydrophobic due to the phase transition, facilitating the release of gas bubbles from the NOPs[82].

Third, the accessibility of the oxygen forming catalysts is reduced in the collapsed state of the gel, due to its reduced volume. Since glucose is consumed by the NOPs, the glucose concentration profile during the process provides a clear demonstration of the gel's accessibility, as depicted in Fig. 6g.

To examine this, samples were taken at specific time intervals in the ascent and descent of the gel, and the glucose concentration of the solution was determined using a UV-Vis-supported glucose assay. The graph clearly illustrates the phases of ascending and descending. Initially, there is a sharp decline in glucose concentration as the shuttle swells and absorbs a significant amount of glucose into the gel. Between the 5th and 10th minute, the critical oxygen concentration in the pores is reached, causing the gel to ascend. As a result of heating, the gel starts to collapse, making it more challenging for glucose to enter the gel and releasing glucose that has not yet been

converted back into the solution. This state persists between the 10th and 40th minute, explaining why the glucose concentration does not decrease further. From the 40th minute onwards, the shuttle descends back into the cooler bottom zone, once again swelling and restoring glucose accessibility. This process subsequently repeats itself, yielding an autonomous and cost-effective material for microplastic degradation (Supplementary Fig. S11).

## Discussion

This work presents a versatile buoyancy-driven material based on functionalized porous organosilica nanoparticles embedded in a thermoresponsive hybrid hydrogel. While previous studies have reported micromotors, photocatalytic systems, and functional hydrogels individually, our approach brings these elements together in a uniquely integrated and autonomous platform. The nanoparticles simultaneously serve as gas reservoirs, photocatalyst carriers, and covalent cross-linkers, enabling self-regulated buoyancy cycles in aqueous media without the need for external fields or magnetic control. The key innovation lies in the deliberate coordination of these functionalities to achieve a self-contained material system capable of independent motion, selective microplastic capture, and spatially localized degradation. The inclusion of a tandem glucose oxidase/Pt-NP catalyst allows efficient in situ generation and storage of oxygen gas within the nanopores, with buoyancy modulated by photothermal heating. Importantly, this system demonstrates the autonomous adsorption, transport, and oxidative degradation of polystyrene microplastics over multiple operational cycles, maintaining its integrity and performance throughout. Looking ahead, the modular design of this platform—enabled by the high degree of functionalizability of the organosilica core—opens new pathways for environmental remediation. By incorporating alternative photosensitizers or adapting the surface chemistry, the BDS-gel concept can be extended to other classes of pollutants (e.g., polyethylene, PET) or coupled with different propulsion mechanisms. To our knowledge, this is among the first examples of a soft material system that demonstrates such a level of functional integration and autonomous environmental response.

## Methods

### General information

Unless stated otherwise, synthesis was performed using general Schlenk techniques under a nitrogen atmosphere. The solvents were dried according to the standard literature[83] and stored under argon. All starting materials used for synthesis were purchased from commercial sources (Merck or ABCR chemicals). The glucose oxidase assay and the hexokinase glucose assay were purchased from Megazyme®. We acknowledge the use of AI-powered tools, including ChatGPT (OpenAI) and DeepL, for assistance in improving the language of this manuscript.

### Preparation of platinum nanoparticles

The platinum nanoparticles were prepared according to literature[55]. A typical synthesis was performed as follows. 56 mg of $Na_2PtCl_4 \cdot 3H_2O$ were dissolved in 22 mL of water before 15 mL of benzyl alcohol were added under vigorous stirring. The reaction mixture was then heated to 80 °C for 12 h. The resulting Pt-NPs were used without further purification.

### Synthesis of 1,3-bis (tri-isopropoxysilyl)−5-styrene

This synthesis has already been published in a previous publication[33]. A typical synthesis was performed as follows. 572 mg (13.5 mmol, 3 eq.) lithium chloride and 157 mg (0.22 mmol, 0.05 eq.) bis(triphenylphosphane)palladium(II) dichloride were dissolved in 60 mL dry dimethylformamide and stirred for 10 min. 2.54 g (4.5 mmol, 1 eq.) of 1,3-bis(tri-iso-propoxysilyl)−5-bromobenzene were added and the

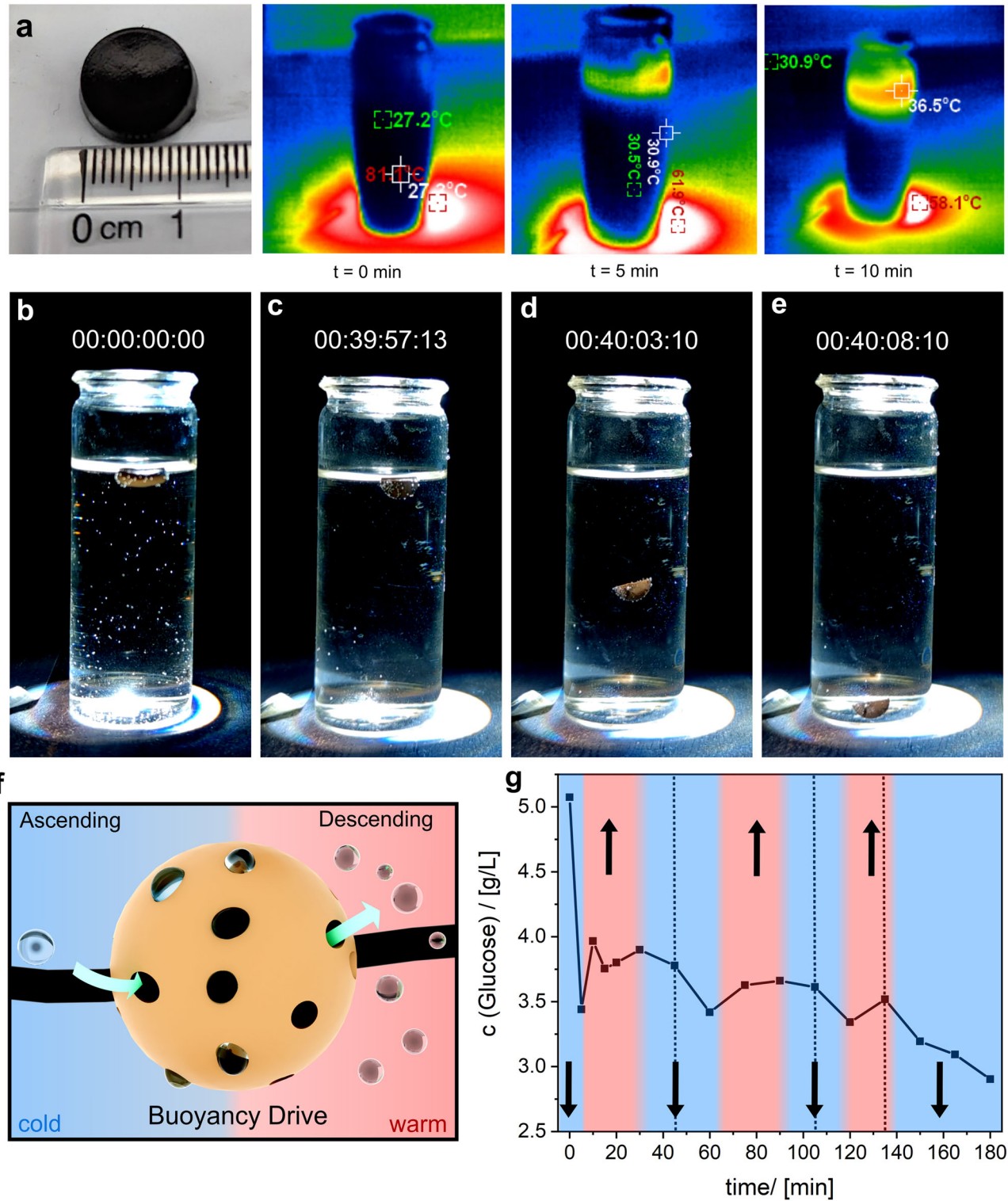

**Fig. 6 | Investigation of the descending process of the BDS-gel when irradiated with a solar simulator. a** Monitoring the photothermal heating of the BDS gel with a thermal imaging camera. **b**–**e** Photographs of the descending process of the BDS gel in a 10 mg/mL D-glucose solution and under irradiation of a sunlight simulator (hh:mm:ss:ms − t₀ starts when the gel has finished ascending – the whole movie process is shown in Supplementary Fig. S10a). **f** Time-resolved glucose concentration of the solution during the irradiation. The red and blue areas represent the ascent (red) and descent (blue) of the BDS-gel. **g** Schematic representation of the temperature-dependent buoyancy behavior of the shuttle gels.

reaction mixture was stirred for further 5 min. 1.7 g (4.5 mmol, 1.2 eq.) of tributyl(vinyl)tin was added dropwise over a period of 5 min before the reaction mixture was heated to 95 °C for 16 h. The reaction was quenched by adding 20 mL water and after cooling the reaction mixture was washed thrice with 50 mL diethyl ether each. The organic phases were combined, dried with magnesium sulfate and the solvent was removed under reduced pressure. The raw product was then purified by ball tube distillation at 110 °C and $5.10^{-2}$ mbar. 1.84 g (3.5 mmol) of 1,3-bis (tri-isopropoxysilyl)−5-styrene were obtained as colorless oil in 96 % yield.

## Synthesis of nanoporous nanoparticles from 1,3-bis (tri-iso-propoxysilyl)−5-styrene

This synthesis has already been published in a previous publication[33]. A typical synthesis was performed as follows. 300 mg (0.6 mmol, 1 equiv.) of the precursor were dissolved in 3.0 mL 2-propanol in a screw cap glass. To initiate hydrolysis 1.86 mL of a 0.01 M hydrochloric acid solution were added under stirring (700 rpm) and the turbid reaction mixture was stirred for 3 h at room temperature (700 rpm). The resulting clear solution was used for the condensation reaction without further purification.

In a screw cap glass 120 mg ($1.8 \cdot 10^{-2}$ mmol) Pluronic® P123, 120 mg (0.33 mmol) of CTAB, 260 μL (1.1 mmol) of 1,3,5-triisopropylbenzene (TIB) and 150 μL (1.6 mmol) of 1-butanol were dissolved in 5 mL of a 0.1 M carbonate buffer (pH = 9.4) for the condensation reaction. The prehydrolyzed precursor species was added in a single shot under vigorous stirring (800 rpm) and the resulting suspension was stirred for 2 d at 25 °C. The colorless solid was separated by centrifugation at $3460 \times g$ and then washed with ethanol, water, and acetone. To remove the organic template, the solid was dispersed in 20 mL of a 1:4 (V:V) solution of concentrated hydrochloric acid and ethanol and stirred for two days at 25 °C. The solid was separated by centrifugation at $3.460 \times g$ and then washed with ethanol, water, and acetone to obtain a colorless powder in 60% yield.

## Introducing Pt-NPs into the pore system of the NOPs

50 mg of vinyl-functionalized NOPs were placed in a Schlenk tube and dried under vacuum overnight. Then, 600 μL of the prepared Pt-NP solution and 100 μL of acetone were carefully added and the particles were stored at room temperature for 24 h before acetone was removed by a constant flow of nitrogen for 20 min. The resulting particles were washed with water and acetone thrice before they were dried in vacuum.

## Glucose oxidase functionalization of the NOPs by thiol-ene click chemistry

50 mg of the functionalized NOPs were placed in a quartz cell. 3 mg (0.01 mmol, 0.05 equiv.) 2,2-dimethoxy-2-phenylacetophenone and 50 mg of glucose oxidase were dissolved in 3 mL of a phosphate buffer (pH = 7.2). The reaction mixture was degassed by nitrogen bubbling before it was irradiated by a mercury vapor lamp for 1 h. The resulting particles were washed with water thrice before they were dried in vacuum. For the activity test, a quinoneimin assay from Megazyme® was used.

## Porphyrin functionalization of the NOPs by thiol-ene click chemistry

To enable thiol-ene-click chemistry with 4,4',4'',4'''-(porphyrin-5,10,15,20-tetrayl)tetrabenzoic acid, a thiol group must first be introduced via one of the carboxyl groups. This was done according to literature[64]. In a general procedure, 250 mg of 4,4',4'',4'''-(porphyrin-5,10,15,20-tetrayl)tetrabenzoic acid were dissolved in 5 mL of $SOCl_2$ in dried glassware. The mixture was refluxed for 4 h at 80 °C under inert atmosphere. The solvent remaining after the reaction was removed under reduced pressure. Subsequently the chlorinated compound was dissolved in 10 mL $CHCl_3$ and the mixture was stirred at 0 °C. Separately, 1 equivalent of cysteamine was dissolved in 10 mL $CHCl_3$ and 2 equivalents $NEt_3$. This solution was then added dropwise to the porphyrin solution. The mixture was covered with aluminum foil and stirred for 2 h. The resulting solution was then washed with $NaHCO_3$ and brine, before drying the organic phase over $MgSO_4$. After evaporation of the organic phase, the resulting amides were obtained as dark purple powder.

For the thiol-ene-click chemistry 50 mg of the functionalized NOPs were placed in a quartz cell. 3 mg (0.01 mmol, 0.05 equiv.) 2,2-dimethoxy-2-phenylacetophenone and 50 mg thiolated porphyrin were dissolved in 3 mL of a phosphate buffer (pH = 7.2). The reaction mixture was degassed by nitrogen bubbling before it was irradiated by a mercury vapor lamp for 1 h. The resulting particles were washed with water and acetone thrice before they were dried in vacuum.

## Hybrid hydrogel synthesis of functionalized NOPs and thermoresponsive pNIPAM

10 mg of functionalized NOPs (Pt-NPs, GOx, Porphyrin) and 100 mg of NIPAM were placed in a screw cap vial and dispersed in 1 mL of distilled water using an ultrasonic bath. 3.0 mg of PPS, 2.7 mg of BIS (N,N'-methylenebisacrylamide) and 8.3 mg of sodium acrylate (SA) were added and the dispersion was degassed by nitrogen bubbling for 10 min before 10 μL of N,N,N,N-tetramethylethylenediamine (TEMED) were added under vigorous stirring. After 30 s, the stirring bar was removed and the dispersion was sealed with a screw cap. The dispersion was left to stand for 12 h before the hybrid hydrogel was extracted from the vial. It was washed by swelling-deswelling cycles (temperature-triggered) in water and ethanol. For the polydopamine coating we used a literature known method[36]. The BDS gel is placed in 20 mL of a 0.01 M TRIS buffer solution before 40 mg of dopamine hydrochloride are added under stirring. After 12 h, the BDS gel was taken out and put into fresh dopamine TRIS-buffer solution under stirring for 12 h until a dark gel was formed.

## Investigating the time until ascending and ascending speed of the hybrid hydrogel shuttle

To show that the time until the shuttle ascends can be controlled by the catalyst concentration, three different NOPs with 0.2 wt%, 0.8 wt% and 1.6 wt% platinum were synthesized. Therefore, 50 mg of vinyl-functionalized NOPs were placed in a Schlenk tube and dried under vacuum overnight. Then, 50 μL (0.2 wt%), 200 μL (0.8 wt%) or 400 μL (1.6 wt%) of the Pt-NP dispersion were carefully added and the remaining volume was filled up to 400 μL with acetone. The particles were stored at room temperature for 24 h before acetone was removed by a constant flow of nitrogen for 20 min. The resulting particles were washed with water and acetone thrice before they were dried in vacuum. Shuttle gels were then prepared according to the synthesis described above. The Pt-NP shuttle gels were then each added to 40 mL of a 0.3 wt% $H_2O_2$-solution and the process was filmed with a camera. The vials were marked every 2 cm to measure the ascending speed of the gels.

## Examination of the ascent and descent process of the hybrid hydrogel shuttle gel

To show that the shuttle system is working, the Pt-NP/GOx/Porph shuttle gel was added to 20 mL of a 1.0 wt% glucose solution and the process was filmed with a camera.

## Microplastic uptake and decomposition using the hybrid hydrogel shuttle gel

For the microplastic uptake tests, 2 μm polystyrene beads were used as a model for microplastic contamination. A 150 μL aliquot of a 10 wt% polystyrene dispersion was diluted with a 3.5 wt% saline solution to a final volume of 15 mL, after which the collapsed shuttle gel was added to the dispersion to allow it to swell. Following the microplastic (MP) uptake, the shuttle gel was extracted from the solution and subjected to a swelling/deswelling cycle in water. To prevent the generation of reactive oxygen species (ROS), the gel was collapsed at 35 °C. The gel was then irradiated for 2 h at 420 nm. After irradiation, a sample was collected and analyzed using thermogravimetric analysis (TGA) to define one complete cycle. This experiment was repeated for three cycles, with the remaining gel collapsed and reintroduced into the original MP solution after each cycle. Additionally, a representative sample was analyzed using Raman spectroscopy and scanning electron microscopy (SEM). To quantify the amount of MP removed from the

solution after each cycle, 1.5 mL aliquots of the polystyrene solution were collected, dried at 80 °C, and weighed.

## Glucose concentration during ascending/descending cycles

A total of 100 mg of the collapsed BDS-gel was dispersed in 40 mL of a 5 mg/mL D-glucose solution, and the vial was irradiated from above using a sunlight simulator. Samples of 1 mL were collected at 5, 10, 15, 20, and 30 min, and subsequently every 15 min until the 3-h mark. Afterward, the D-glucose concentration was determined using a hexokinase assay (Megazyme®) in combination with UV-Vis spectrophotometry.

## Analytical methods

ATR-IR Spectroscopy was performed on a Bruker Tensor with an ATR unit. All spectra were background corrected and normalized to the Si-O-Si vibration at 1044 cm$^{-1}$. Raman spectroscopy was performed on a Bruker Senterra. $N_2$-physisorption measurements were performed with the Micromeritics Tristar 3020 at −196 °C. Prior to analysis the samples were degassed at 85 °C for 720 min. Thermogravimetric Analysis measurements were performed using a Netzsch STA 409 PC LUXX with a heating rate of 5 K/min until 300 °C and 10 K/min for 300–1000 °C. NMR measurements ($^1$H, $^{13}$C) were performed on a Bruker Ascend 400 MHz spectrometer. Scanning electron microscopy was performed on a Hitachi Regulus SU8200. UV-Vis spectroscopy was performed on a Agilent UV-Vis Cary 4000. The DLS measurements were done by using a Malvern Zen5600. TEM was acquired on a FEI Tecnia G2 F20 TMP. A 300 W xenon lamp with an AM 1.5 G filter (LS0308 from Quantum Design) was used to simulate sunlight irradiation. The light intensity is 1 Sun (100 mW/cm$^2$).

## Data availability

The data supporting the findings are provided within this Article and its Supplementary Information. Extra data are available from the corresponding author upon request. Source data are provided with this paper.

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

## Acknowledgements

All analytical measurements were performed in the central analytical facility cfMATCH. We thank Jakob Schlenkrich (Institute of physical chemistry and electrochemistry, LUH Hannover) for providing his expertise in the solar simulator setup.

## Author contributions

D.K.: conceptualization, data curation, formal analysis, investigation, writing-original draft. F.K.: synthesis of the materials. S.R.: Synthesis of the materials. Y.K.: conceptualization, supervision, verification, writing-original draft. S.P.: supervision, funding acquisition, writing-original draft, project administration.

## Funding

## Competing interests

The authors declare no competing interests.
