## [Transparent Peer Review file · Nature Communications]

A Self-Regulating Shuttle for Autonomous Seek and Destroy of Microplastics from Wastewater

Corresponding Author: Professor Sebastian Polarz

Version 0:

Reviewer comments:

Reviewer #1

(Remarks to the Author)

Title: A Self-Regulating Shuttle for Autonomous Seek and Destroy of Microplastics from Wastewater (NCOMMS-24-36108-T)

Comments:

I am not able to recommend this manuscript for Nat Comm as I don't see any novelty here.

1. Several works are going on microplastic adsorption and degradation using hydrogel, for example, DOI: 10.1039/D3NR06115A, 10.1021/acs.chemmater.2c00625. What is the novelty of this work? How this will contribute to the existing literature of this field?
2. The authors created a buoyancy-driven shuttle for microplastic (MP) removal, where a gel is coated with a black polydopamine layer. However, both polydopamine and common microplastics carry negative charges. Given this, how can microplastics efficiently adsorb onto the hydrogel? The mechanism appears very weak due to electrostatic repulsion.
3. The MP removal mechanism needs a thorough explanation. At the moment, seems like a hypothesis.
4. What factors influence material stability and MP adsorption efficiency in water? How do the varying water pH levels affect removal efficiency? In which real-world applications, like seawater, drinking water, or wastewater, will these materials be employed?
5. Authors have discussed that the photocatalyst (porphyrin) in the NPs produces ROS, breaking down microplastics at the center. Additionally, the gel's black color causes heating. How effective is this black color alone in inducing microplastic degradation? Also, an analysis is needed on the cytotoxicity of porphyrin. Too many variables.
6. How stable the created hydrogel is in terms of its mechanical and chemical characteristics, described in detail using sufficient experimental evidence.
7. How the author performed microplastic detection and quantification. clarify
8. What is the reusability test of the hydrogel for MP removal? Is there any possibility of leaching out the adsorbed MPs from the hydrogel matrix in its swell state?
9. The photo-degradation of microplastics can lead to the formation of nanoplastics through various processes. These nanoplastics have the potential to recontaminate water sources, posing ongoing environmental concerns. How to resolve this issue?
10. The effectiveness of this prepared shuttle in bouncing is notable, yet its efficacy in removing MPs from real-water applications is limited. A comprehensive experimental analysis, along with thorough materials characterization under varied conditions, is crucial to substantiate these observations.

There are several points that need additional experiments or discussion and it's not ready yet for Nat Comm.

Reviewer #2

(Remarks to the Author)

Microplastic pollution has aroused more attention in recent years, and it is urgent to know how to remove microplastic from both aquatic and soil environment. The manuscript of "A Self-Regulating Shuttle for Autonomous Seek and Destroy of Microplastics from Wastewater" is interesting, and indicated an innovation for both theory and technique on how to remove the microplastic from the water. However, there are several issues needed to be concerned before acceptance;

(1) Did authors want to remove the microplastic from the water or sediment, and fresh water or sea water, or effluent? The scope of this method used should be mentioned in both introduction and methods, even in the abstract. As well, simulation of natural environmental situation should be also considered in this study.

(2) The swelling status was only seen in the figures, but was not found in the videos. The process of descending and ascending were not smooth in the videos, and maybe the structure or the morphology of shuttle needs to be improved?

(3) Only the microplastic in the gel was measured, while the its concentration dynamics in solution was not measured. This could misjudge the results, and microplastic could escape from gel during the washing process.

(4) Why the author didn't use this method/technique to finish a whole process (several cycles) including of swelling-uptake, ascending, floatation-decomposition(microplastic)-shrinking, and then detect the microplastics in both water and gel? This is the best way to test the efficiency of this method on microplastic removal.

(5) It is a big issue that the buoyancy is driven by the D-glucose oxidation, and this leads to the method could be only used in effluent treatment and the cost could be also high. As well, there are many kinds of pollutants including both particles and water dissolve substances in aquatic environment, and this could deposit on the surface of gel, even block the pores inside. Therefore, authors needs to address these issues.

(6) Why was only the polystyrene tested in this study? How about the PE, PP, PET, PVC?

(7) The "thermally switchable buoyancy" could be not working in the natural water situation when the temperature does not satisfy the need in the surface or bottom of the water.

(8) Did author also consider the types, density, size and shape of microplastics? Can the ROS-induced decompose all microplastics? Is the time costed of microplastic decomposition same to gel floatation?

Several specific comments

(1) Section of Main

Repeated words is suspected in the sentence of "The high stability and low biodegradability....."

(2) Scheme 1. The full name of nanoporous organosilica particles mentioned once is enough, and the abbreviation of NOPS should be used in the next. Please check the whole manuscript, and avoid the similar issues.

Reviewer #3

(Remarks to the Author)

This manuscript presents a multiresponsive hydrogel shuttle for microplastic removal from water. Notably, the hydrogel scaffold's combination of various functional units enables multiple responses for microplastic capture. This composite structure resembles a 3D micromotor capable of capturing a substantial amount of microplastics, albeit requiring additional fuel, such as a high-concentration glucose solution, to propel its motion. The hydrogel's descending motion is further facilitated by photothermal shrinkage of the temperature-responsive scaffold. Despite an intriguing concept, the complex stimuli required to activate and drive the hydrogel shuttle hinder its practical field application. Moreover, the manuscript lacks a continuous demonstration of the proposed shuttle (as depicted in Scheme 1). Furthermore, the claim of microplastic degradation through photocatalytic ROS generation is highly questionable, considering the well-established challenges in photocatalytically decomposing microplastics (e.g., *Environ. Sci. Technol.* 37, 2003, 4494 using TiO₂ and Cu-phthalocyanine photocatalyst). Additionally, the manuscript contains numerous grammatical errors and is written in a conversational style, impeding evaluation. Overall, this manuscript appears unsuitable for publication in *Nature Communications*. The following are additional comments for the authors' future submission elsewhere:

1. The authors claim that ROS generated by the tethered porphyrin unit can fully degrade PS microplastics. However, the degradation of rigid PS microplastics through porphyrin-based photocatalytic treatment is highly doubtful. The photothermal effect of porphyrin units may loosen the hydrogel-microplastic interaction, leading to desorption. To confirm microplastic photocatalytic decomposition, control experiments are necessary: 1) exposing a PS microplastic solution with the porphyrin unit to UV light and 2) detecting corresponding decomposition residues (e.g., CO₂ concentration in headspace, TOC in water, and microplastic mass change in the solution).

2. The authors claim hydrogel scaffold stability during experiments, yet the provided structural analysis is insufficient. XPS, ICP (for Pt), and SEM analyses on the cycled hydrogel, including quantitative analysis of tethered Pt, PNIPAM, and porphyrin units, should be included.

3. The mechanism by which the PDA coating prevents hydrogel scaffold degradation is unclear. The intact hydrogel structure (as claimed by the authors) might indicate microplastic detachment rather than decomposition.

4. Figure S1f: Please clarify how the hydrogel is regenerated through water washing.

5. Figure 3: Ensure the figure description accurately matches the figure number.

Version 1:

Reviewer comments:

Reviewer #1

(Remarks to the Author)

1. Define the abbreviation 'pNIPAM' in full within the abstract.

2. Present a plot of zeta potential versus pH for both BDS Gel and PS beads.

3. Please provide the SEM image of the BDS hydrogel after MP uptake, maintaining the same scale as the pre-uptake images.

4. Provide an estimate of the operational costs associated with MP removal using the BDS hydrogel.

5. Clarify the atomic weight percentage of Pt, considering its minimal intensity in Fig. S5(k).

6. Rectify typographical errors found in the figure captions of the Supplementary Information.
7. It would be good if the author could add the scale bar inside the TEM and SEM images instead of writing as a figure caption.

Reviewer #2

(Remarks to the Author)

This paper has been carefully addressed according to reviewers' comments, and current version is suitable for publication.

Reviewer #3

(Remarks to the Author)

While the authors have addressed some of the previous concerns, the manuscript, in its current state, does not demonstrate sufficient novelty for publication in *Nat. Commun.* The concept of fuel-driven micromotors is well-established, and the proposed innovation, a descending motion achieved through photothermal shrinkage of a temperature-responsive scaffold, lacks robust experimental validation. Specifically, the authors fail to provide compelling evidence of continuous shuttle operation, as depicted in Scheme 1, casting doubt on the practical viability of their concept. Furthermore, the claim of PS photodegradation by the reported materials remains highly questionable. While presented as a proof-of-concept, the presented data is insufficient to support this assertion. The new TGA and DLS analyses are inadequate for characterizing the comprehensive decomposition of PS. Detecting the degradation of 15 mg of PS within a 150 μ L, 10 wt% solution using these techniques is inherently challenging. Moreover, typical DLS analyses of microplastic degradation demonstrate surface weathering and gradual size reduction during photocatalytic treatment (*J. Environ. Chem. Eng.* 2022, 10, 108195). The current study's photodegradation experiments, conducted over a mere two hours under 420 nm irradiation, are insufficient to achieve significant PS degradation, despite the authors' claims regarding singlet oxygen generation. The inherent difficulty of microplastic degradation in aqueous environments is well-documented, as evidenced by numerous studies, including *Environ. Sci. Technol.* 37, 2003, *ACS Sustain. Chem. Eng.* 2023, 11, 10688, *ChemSusChem* 2024, 17, e202301350, *Solar RRL* 2023, 7, 2300411, and many more. To strengthen their claims, the authors should conduct degradation tests using standardized, dyed PS microplastic dispersions, which would allow for direct visual confirmation of microplastic fate. Additionally, incorporating photocatalytic degradation experiments using model organic pollutants (e.g., dyes) would provide valuable insights into the efficiency of the developed materials. Compounding these scientific concerns is the persistent lack of improvement in the manuscript's clarity and language, including the presence of typographical errors within highlighted revised sections. This further detracts from the overall quality of the work. In conclusion, the current manuscript lacks the necessary novelty, experimental rigor, and supporting data to warrant publication in *Nat. Commun.*

Version 2:

Reviewer comments:

Reviewer #1

(Remarks to the Author)

All the queries have been resolved.

Point to Point answer to our Manuscript (NCOMMS-24-36108-T)

"Comments given by the reviewers"

and **our answers** (*, the page and line number given herein are regarding the marked manuscript)

REVIEW 1

(1.1) *"I am not able to recommend this manuscript for Nat Comm as I don't see any novelty here. Several works are going on microplastic adsorption and degradation using hydrogel, for example, DOI: 10.1039/D3NR06115A, 10.1021/acs.chemmater.2c00625. What is the novelty of this work? How this will contribute to the existing literature of this field?"*

Our answer: We thank the reviewer for the attentive reading of the manuscript and the valuable feedback, which showed us that we had to improve clarity of the manuscript.

The primary innovation of our study lies in the development of a truly autonomous system capable of cyclic ascent and descent, driven by the reversible capture and release of gas within the pore system of nanoporous organosilica particles. This unique mechanism provides significant advances in hydrogel behaviour, which has not been demonstrated previously in the literature. Furthermore, while microplastic degradation serves as a proof-of-concept application, it is not the sole focus of our study. Our system provides a highly adaptable platform that can be readily tailored for various environmental challenges. The exceptional flexibility of the silica nanoparticles allows for straightforward functionalization (e.g., tuning hydrophilic/hydrophobic properties, integrating alternative ROS producers, or incorporating specific enzymes), enabling rapid adaptation to different target pollutants and operational conditions. **We therefore have revised the whole manuscript to more clearly highlight these aspects**, ensuring that the novelty and broader applicability of our work are fully conveyed (**e.g. page 2 line 15-22; page 4 line 64-66; page 7 scheme 2; page 20 line 334 to page 21 line 343**). We appreciate the reviewer's insightful comments, which have helped us refine our presentation.

(1.2) *"The authors created a buoyancy-driven shuttle for microplastic (MP) removal, where a gel is coated with a black polydopamine layer. However, both polydopamine and common microplastics carry negative charges. Given this, how can microplastics efficiently adsorb onto the hydrogel? The mechanism appears very weak due to electrostatic repulsion."*

Our answer: We thank the reviewer for highlighting this important detail, as it shows that we haven't made this point clear enough. **Therefore, we added further explanations and literature references to the manuscript (page 8 line 107-112; reference 42-44) and also overworked Figure 1 (page 9)**. While it is correct that both polydopamine and most common microplastics (e.g., polystyrene) carry negative charges under typical environmental conditions, the adsorption mechanism is not primarily driven by electrostatic interactions. Instead, the strong adhesive properties of

the PDA coating play a crucial role in facilitating microplastic uptake. Specifically, the polydopamine layer lowers the zeta potential of the hydrogel surface, which modifies the overall surface charge and enhances its “glue-like” interaction capabilities. Moreover, π - π stacking interactions between the aromatic structures of polydopamine and polystyrene contribute significantly to the adsorption process. These non-covalent interactions are well-known to dominate in systems involving PDA coatings and organic contaminants, providing an effective and robust mechanism for capturing microplastics.

(1.3) *“The MP removal mechanism needs a thorough explanation. At the moment, seems like a hypothesis.”*

Our answer: We thank the reviewer for highlighting this important detail. We fully agree with the reviewer that the MP removal mechanism needed more explanation in the manuscript. **We therefore revised the manuscript and added further details and literature (page 14, line 216-228, references: 25,54-58, 60-65) as well as additional experiments (page 15, Figure 3g; Supporting information: page 24, Figure S5h).** While the exact mechanism of microplastic (MP) removal is still under ongoing investigation in the current literature, it is believed that the reaction pathways primarily involve the addition of reactive oxygen species, carbon-carbon scissions, and hydrogen abstractions. Our hypothesis suggests that the (rather hydrophobic) degradation products remain within the hydrogel aids in the further decomposition process, as this increases the contact time with ROS, thereby facilitating decomposition into at least aromatic oxygenates or CO₂. In support of this, our previous publication demonstrated the retention of organic substances within the hydrogel (**Reference 32**). Additionally, **we conducted Dynamic Light Scattering (DLS) experiments** after UV irradiation, which revealed no detectable microplastic particles outside the hydrogel (**Supporting information: page 24, Figure S5h**). Given that DLS can detect particles as small as a few nanometers, we do not believe nanoplastics are formed. From Thermogravimetric analysis it can be concluded that a nearly complete decomposition of the microplastics within the gel occurs. Furthermore, experiments involving a gel without a photosensitizer showed that the microplastic remained inside the hybrid hydrogel, providing additional evidence that ROS are likely involved in the decomposition process. While we cannot definitively prove complete degradation to CO₂, these results demonstrate that the microplastic is no longer present within the gel after irradiation, but also no degradation byproducts can be found in solution. This suggests that the microplastic has undergone at least partial decomposition to aromatic oxygenates. **We revised the manuscript accordingly (page 16 line 254 to page 17 line 277)**

- (1.4) *“What factors influence material stability and MP adsorption efficiency in water? How do the varying water pH levels affect removal efficiency? In which real-world applications, like seawater, drinking water, or wastewater, will these materials be employed?”*

Our answer: We thank the reviewer for highlighting this important aspect as it shows that we could improve the clarity of the manuscript. While our manuscript primarily focuses on microplastic decomposition as a proof of principle for the system’s applicability, we appreciate the opportunity to address the factors influencing material stability and microplastic adsorption efficiency. The general material stability of the hydrogel has been characterized in our latest publication (**Reference 32**). Due to the presence of negative charges from the acrylate groups introduced during gel synthesis, the transition from the swollen to the collapsed state requires some salt or acid to shield or protonate the carboxylate groups. Therefore, we would recommend the gel’s use in seawater, where natural salt concentrations can help facilitate this process. Additionally, the glucose oxidase enzyme, which is crucial for the operation of this specific system, remains active under moderate pH conditions ranging from pH 4 to 10, which covers the pH range typically found in seawater. **We added the reference to the manuscript (page 10, line 154-156; reference 51)**. Thus, seawater would be an ideal environment for this system. The removal efficiency itself should not be significantly influenced by the pH value in the expected pH range (pH 5-8), since neither polydopamine, nor the photosensitizer should undergo any changes in this regime. **We added the literature references to the manuscript (page 8, line 107-112, reference 41)**

- (1.5) *“Authors have discussed that the photocatalyst (porphyrin) in the NOPs produces ROS, breaking down microplastics at the center. Additionally, the gel’s black color causes heating. How effective is this black color alone in inducing microplastic degradation? Also, an analysis is needed on the cytotoxicity of porphyrin. Too many variables.”*

Our answer: We thank the reviewer for the constructive feedback as it shows that we could improve clarity for future readers. **Therefore, the manuscript was revised by a more detailed description of the reference experiments we performed (page 14 line 235 to page 15, line 239; page 16 line 248-250; line 254-267; page 17, line 268-277)**. To confirm that microplastic degradation was caused by reactive oxygen species (ROS) rather than heat generated by the polydopamine coating, we conducted a microplastic loading and degradation experiment using two otherwise identical BDS gels, with one lacking the photosensitizer. After the degradation process, the BDS gels were analyzed using thermogravimetric analysis (Figure 3f). The results demonstrate that negligible microplastic degradation occurs in the absence of the photosensitizer,

leading to the conclusion that thermal heating alone does not contribute to microplastic degradation.

Furthermore, we would like to thank the reviewer for their comment regarding the cytotoxicity of porphyrins. While conducting our own study on this topic would exceed the scope of the current publication, a considerable amount of research on the cytotoxicity of porphyrins is already available in the literature. **We revised the manuscript in this regard and added additional literature (page 14, line 227-228; reference: 60-65).** Due to their ability to generate ROS, porphyrins are considered cytotoxic under irradiation, as the resulting ROS reacts non-selectively with its immediate surroundings. However, the range of these interactions is very limited. In contrast, porphyrins are generally considered to have low cytotoxicity in the absence of irradiation. For this reason, porphyrins are already being used in anti-cancer photodynamic therapy, where they are employed to selectively destroy cancer cells in the body with ROS support, without damaging healthy cells. Based on this, we believe it is reasonable to assume that porphyrins are unlikely to have a negative impact on marine life.

- (1.6) *“How stable the created hydrogel is in terms of its mechanical and chemical characteristics, described in detail using sufficient experimental evidence.”*

Our answer: We thank the reviewers for bringing this unclarity to our attention.

While we did not explicitly demonstrate the mechanical stability of the gels with experimental data in the current manuscript, **we would like to refer to our publication from 2023 (reference 32).** In that study, we developed a similar hybrid hydrogel composed of vinyl-functionalized organosilica nanoparticles (NOPs) and pNIPAM, which was fully characterized. Nanoindentation measurements revealed that the mechanical stability of the gels benefits from the additional crosslinking provided by the silica particles, resulting in materials that are robust and easy to handle.

Regarding the chemical stability of the gels (against ROS), we agree with the reviewer that the experimental evidence could be improved. Therefore, we have performed additional experiments and updated the manuscript accordingly (page 17, lines 278–285; Supporting Information, Figure S5k,l). EDX measurements verified that platinum, silicon, and sulfur (originating from porphyrin and GOx) remained present in the cycled hydrogel, confirming the stability of the scaffold. Moreover, SEM imaging showed no visible differences before and after irradiation, while MAS solid-state NMR analysis indicated that the hydrogel composition remained unchanged throughout cycling. Along with the infrared spectroscopy data (**Supporting Information, Figure S5i**), these findings collectively demonstrate the robustness of the

shuttle gel and its capacity for repeated use in microplastic degradation. We also evaluated the microplastic loading and decomposition over three full cycles (**page 15, Figure 3g; supporting information, Figure S5j**). Microplastic detection was performed using thermogravimetric analysis before and after microplastic loading, as well as after solar irradiation. The results show that the hybrid hydrogel maintains both its capacity and removal efficiency across multiple cycles, with no significant reduction in performance. We believe these additional analyses provide strong evidence of the hydrogel's stability during the experiments and appreciate the reviewer's insightful feedback, which has helped us improve the manuscript.

(1.7) *“How the author performed microplastic detection and quantification. Clarify”*

Our answer: We agree with the reviewers' recommendation and clarified the whole section dealing with microplastic detection and quantification in the manuscript (**page 14 line 235 to page 17 line 277**) and the Supporting Information (**page 24, Figure S5h; page 25, Figure 5j**). Microplastic detection was primarily performed using scanning electron microscopy (SEM) to visualize the presence or absence of microplastics on the gel surfaces. Quantification of microplastic degradation was achieved through thermogravimetric analysis (TGA), which allowed us to determine the percentage of microplastic weight loss after treatment. Our results demonstrated a significant reduction in microplastic content, with the porphyrin-functionalized NOPs showing a 98.7 wt% degradation compared to the 3.2 wt% observed in the control (without ROS producent). Additionally, Raman spectroscopy was used to confirm the presence of specific microplastic vibrational bands, such as those corresponding to polystyrene, before and after irradiation, further supporting the success of microplastic decomposition by reactive oxygen species (ROS) (**Supporting Information, page 24, Figure S5h**). To rule out the possibility that small fragments, such as nanoplastics, leave the hydrogel, **Dynamic Light Scattering (DLS) measurements were performed** on the washing solution **after the decomposition process, and no significant signal indicative of nanoplastics was detected** (**Supporting Information, page 24, Figure S5h**).

(1.8) *“What is the reusability test of the hydrogel for MP removal? Is there any possibility of leaching out the adsorbed MPs from the hydrogel matrix in its swell state?”*

Our answer: We appreciate the reviewer's constructive feedback and agree that further demonstration of the BDS-gel's reusability is essential. **To address this, we**

have conducted additional experiments, now included in the main manuscript (page 15, Figure 3g; page 17, lines 278 to page 18, line 294) and the Supporting Information (page 24, Figure S5h; page 25, Figure S5j).

We evaluated the microplastic loading and decomposition over three full cycles. Microplastic detection was performed using thermogravimetric analysis before and after microplastic loading, as well as after solar irradiation. The results show that the hybrid hydrogel maintains both its capacity and removal efficiency across multiple cycles, with no significant reduction in performance. Furthermore, we assessed potential microplastic loss from the hydrogel by comparing the microplastic content in the gel (determined by TGA) with the amount remaining in solution (determined by drying and weighing the residue). Our findings indicate that the microplastic removal process is consistent and reproducible, with the amount of microplastic leaving the solution closely matching the amount retained by the BDS-gel. Additionally, dynamic light scattering measurements confirm that no microplastic particles or fragments leach from the gel during irradiation (**Supporting Information, Figure S5h**). These results further support the stability and reusability of the hydrogel for microplastic removal over multiple cycles.

Regarding the possibility of MP leaching from the hydrogel matrix in its swollen state, we conducted control experiments to assess this risk. Dynamic light scattering (DLS) experiments confirmed that no micro- or nanoplastic particles escaped from the hydrogel during the process (**Supporting Information, page 24, Fig. S5h**). Furthermore, we investigated MP mass changes by drying the solution and weighing the remaining MPs (**Supporting Information, page 25, Fig. S5j**). The results were in good agreement with thermogravimetric analysis of MP uptake by the hydrogel, further supporting the conclusion that leaching is negligible.

(1.9) *“The photo-degradation of microplastics can lead to the formation of nanoplastics through various processes. These nanoplastics have the potential to recontaminate water sources, posing ongoing environmental concerns. How to resolve this issue?”*

Our answer: We thank the reviewer for highlighting this important detail. We fully agree with the reviewer that the formation of nanoplastics through the photo-degradation of microplastics is an important environmental concern.

Regarding the specific issue of nanoplastic formation in our study, we have not observed any nanoplastics in our current analysis. To further analyse this, **we conducted Dynamic Light Scattering (DLS) measurements** of the washing solution after the decomposition process, **and no significant signal indicative of**

nanoplastics was detected (page 16, line 263-266; Supporting information, page 24 Figure S5h). Additionally, in our Raman spectroscopy analysis, no signals related to the formation of nanoplastics were found (**Supporting information, page 23 Figure S5g**). Based on these findings, we hypothesize that the more hydrophobic nature of the BDS-gels, compared to the surrounding water, may result in the plastics remaining longer within the gel. This could allow for more complete decomposition of the microplastics (to aromatic oxygenates or CO₂), rather than leading to the formation of nanoplastics. However, we recognize that this is currently a hypothesis.

(1.10) *“The effectiveness of this prepared shuttle in bouncing is notable, yet its efficacy in removing MPs from real-water applications is limited. A comprehensive experimental analysis, along with thorough materials characterization under varied conditions, is crucial to substantiate these observations.”*

Our answer: We thank the reviewer for his thoughtful feedback and suggestions. While we greatly appreciate the recommendation to conduct additional experiments under varied conditions for real water applications, we believe that implementing these would significantly extend the scope of the current manuscript. As the primary focus of this study was to describe the autonomous ascending and descending movement by our BDS-gel, which can be used to decompose microplastic on a proof of principle level, incorporating these new experiments would require a substantial amount of additional work and could shift the direction of the paper beyond its intended scope.

However, we have taken the reviewers feedback into account and have conducted additional structural analyses and revised the manuscript accordingly (page 16, line 263 to page 18, line 294; Supporting Information, Figure S5k,l). Specifically, EDX measurements confirmed the continued presence of platinum, silicon, and sulfur (from porphyrin and GOx) in the cycled hydrogel, demonstrating the stability of the scaffold. Moreover, SEM images showed no optical differences before and after irradiation, while MAS solid-state NMR analysis revealed no compositional changes upon cycling. Together with the infrared spectroscopy data (**Supporting Information, Figure S5i**), these results confirm the structural integrity of the shuttle gel and its robustness for repeated use in the microplastic degradation process. Also, we could show the reusability of the BDS-gel over three full cycles (**page 15, Figure 3g; supporting information page 25, Figure S5j**). We hope that these revisions sufficiently address your concerns and strengthen the overall manuscript.

(1.11) *“There are several points that need additional experiments or discussion and it's not ready yet for Nat Comm.”*

Our answer: The comments have helped elevate the manuscript to another level, and we truly appreciate the time and effort dedicated to improving our work. **We have carefully addressed all concerns through additional experiments and discussions, which have further strengthened the manuscript.** We hope that with these revisions, the manuscript is now suitable for consideration in Nature Communications.

REVIEW 2

(2.1) *“Microplastic pollution has aroused more attention in recent years, and it is urgent to know how to remove microplastic from both aquatic and soil environment. The manuscript of “A Self-Regulating Shuttle for Autonomous Seek and Destroy of Microplastics from Wastewater” is interesting, and indicated an innovation for both theory and technique on how to remove the microplastic from the water. However, there are several issues needed to be concerned before acceptance”*

Our answer: We thank the reviewer for the positive feedback on our manuscript and for recognizing the novelty and importance of our work on microplastic removal from wastewater. We are committed to addressing the reviewers concerns to further improve our manuscript.

(2.2) *“Did authors want to remove the microplastic from the water or sediment, and fresh water or sea water, or effluent? The scope of this method used should be mentioned in both introduction and methods, even in the abstract. As well, simulation of natural environmental situation should be also considered in this study.”*

Our answer: We thank the reviewer for highlighting this important detail. We fully agree with the reviewer that we had to improve clarity in this point. Therefore, we have revised the manuscript accordingly to clarify the focus of our study (**page 2, line 15-19 and line 21-22; page 4, line 64-66; page 7, Scheme 2; page 20 line 334 to page 21 line 343**). The primary focus of our work was the demonstration of the autonomous ascending and descending movement of the BDS gel. The microplastic decomposition observed in this study serves as an initial application of these systems. While this is an important step, the key aim was to showcase the gel's capability for autonomous motion. Microplastic decomposition, therefore, represents a proof of concept for the broader potential applications of this technology. Regarding the environmental context, we specifically targeted seawater for our experiments, as the thermoresponsive switching of the gel requires a small amount of salt to shield the high charge density within the gel. **To improve clarity in this point, we revised the manuscript (page 8, line 126-**

127 to page 9 line 129) and the Supporting Information (page 4, line 101) accordingly. The Microplastic is then collected in the water layer next to the seabed. Additionally, when the BDS-gel descends, a little bit of sediment is stirred up and taken up by the gel. While this initial approach centers on seawater, we acknowledge the importance of simulating more complex environmental conditions, such as interactions with water or sediment matrices

(2.3) *“The swelling status was only seen in the figures, but was not found in the videos. The process of descending and ascending were not smooth in the videos, and maybe the structure or the morphology of shuttle needs to be improved?”*

Our answer: We thank the reviewer for the constructive feedback as this comment has made us realize the importance of more clearly emphasizing this aspect in the manuscript to prevent any potential misunderstandings. The swelling behaviour shown in Figure 1b represents the maximum swelling after 3 hours. In contrast, the gel's ascent in the video occurs after just 9 minutes, so the swelling is not as visibly pronounced at this stage. However, during descending, the process is more clearly observable in the video when comparing the size of the gel at $t=0$ min and after 40 minutes. The delayed collapse (40 minutes in the video versus 10 minutes in Figure 1b) can be explained by the different heating methods used. In Figure 1b, the entire solution containing the gel was heated, whereas in the video, the heating was achieved only through irradiation, with the surrounding water initially cooling the gel. **We re-rendered the video to improve its smoothness (Supporting Information Video S1).** We have already demonstrated that the ascent time can be adjusted by modifying the Platinum-content in the pores of the NOPs. A longer time till ascending will also optimize the swelling properties. However, since the gel can undergo multiple cycles, we do not consider this optimization to be critical for the current study. Regarding the optimization of the gel's shape, we agree that this is an interesting suggestion. We chose the cylindrical shape because it offers a good surface-to-volume ratio while maintaining mechanical stability. A flat square design could indeed increase the irradiated surface area, and it should be synthetically feasible. However, we believe that exploring this idea would go beyond the scope of the current paper.

(2.4) *“Only the microplastic in the gel was measured, while the its concentration dynamics in solution was not measured. This could misjudge the results, and microplastic could escape from gel during the washing process.”*

Our answer: We agree with the reviewer's recommendation to also investigate the microplastics present in the solution. Since we do not have access to a static light scattering setup to measure the concentration of microplastics in the solution, **we determined the weight difference of the microplastics before and after uptake by the BDS-gel (Supporting information, page 25, Figure S5j)**. These results align well with those obtained from thermogravimetric analysis. To ensure no microplastics escape during the washing process, **we conducted dynamic light scattering measurements on the washing solution (page 16, line 263-266; Supporting Information, page 24, Figure S5h)**. As we detected no signals in the washing solution and the thermogravimetric analysis of the BDS-gel before and after washing showed no reduction in microplastic content, **we are confident that no microplastic escape occurs during washing. To enhance clarity, we have moved the thermogravimetric analysis results, which were originally in the Supporting Information, to the main manuscript (Figure 3f)**.

- (2.5) *“Why the author didn’t use this method/technique to finish a whole process (several cycles) including of swelling-uptake, ascending, floatation-decomposition(microplastic)-shrinking, and then detect the microplastics in both water and gel? This is the best way to test the efficiency of this method on microplastic removal.”*

Our answer: We appreciate the reviewer’s constructive feedback and fully agree that demonstrating the complete cyclization of the BDS-gel is essential. **To address this, we repeated the experiment for three full cycles.** After each cycle, a sample was collected to quantify the microplastic content in the gel by thermogravimetric analysis. Additionally, we conducted an experiment comparing the microplastic content in the solution versus that in the gel. Therefore, the amount of microplastic in the Gel was determined by thermogravimetric analysis, while the amount of microplastic in solution was determined by drying the solution and weighing the residue. **Our findings show that the gel maintains its capacity for microplastic uptake across multiple cycles without a reduction in performance. Also, the amount of microplastic leaving the solution fits well with the amount of microplastic in the BDS-gel. The results of these experiments have been incorporated into the manuscript (page 15, Figure 3g; page 17 to page 18 line 294) and the Supporting Information (page 25, Figure S5j).**

- (2.6) *“It is a big issue that the buoyancy is driven by the D-glucose oxidation, and this leads to the method could be only used in effluent treatment and the cost could be also high. As well, there are many kinds of pollutants including both particles and water dissolve substances in aquatic environment, and this could deposit on the surface of gel, even block the pores inside. Therefore, authors needs to address these issues.”*

Our answer: We thank the reviewer for highlighting this important detail. We fully agree with the reviewer that the usage of D-glucose oxidase is not ideal for an effective application in different water types. However, we believe that a low D-glucose concentration does not affect the overall ascent mechanism but rather influences the time required for the gel to rise. The current version of the BDS-gel is primarily designed as a model system to demonstrate the concept of reversible gas storage within the pores of NOPs, functioning as a buoyancy-driven shuttle. **We have revised the manuscript accordingly to clarify the focus of our study (page 2, line 15-19 and line 21-22; page 4, line 64-66; page 7, Scheme 2; page 20 line 334 to page 21 line 343).** Given the high flexibility in functionalizing the NOPs, alternative driving mechanisms could certainly be explored in the future. One particularly elegant approach could involve utilizing the CO₂ released during the degradation of microplastics as the driving force. However, this would require ROS production in the dark (on the seabed), which has proven to be quite challenging. Developing an alternative propulsion system will certainly be part of our future research efforts. For the scope of this manuscript, however, introducing a new propulsion system would be beyond the intended focus and too extensive for this work.

We appreciate the reviewer's insightful comment regarding potential pore blocking by various substances present in natural water bodies. The reviewer is correct that real aquatic environments contain a wide range of dissolved and particulate pollutants that could potentially affect the hydrogel's performance. However, literature reports have demonstrated that hydrogels can be successfully applied in real water systems without significant loss of functionality due to pore blocking (e.g., Priestley et al., **Reference 36**). Furthermore, in our previous study (**Reference 32**), we showed that organic contaminants, specifically the dyes Solvent Yellow 14 and Methylene Blue, did not negatively impact the properties of the hybrid hydrogel. These findings suggest that the material maintains its functionality even in the presence of organic pollutants.

(2.7) *“Why was only the polystyrene tested in this study? How about the PE, PP, PET, PVC?”*

Our answer: We thank the reviewer for the feedback and agree that a comprehensive study involving various types of microplastics would be valuable. **Our decision to initially focus on the degradation of polystyrene was based on two key reasons.** Firstly, **polystyrene is one of the most prevalent microplastics found in marine pollution.** Polystyrene is the third microplastic in terms of global waste generation after PP and PE, with 17 million tonnes per year. Given that Polystyrene microplastic is the

only one of those three candidates with a density higher than seawater it makes it an ideal model candidate for this study. Secondly, studies suggest that **polystyrene**, with its relatively acidic tertiary hydrogen atom, **is particularly susceptible to attack and degradation by singlet oxygen**. Singlet oxygen is the predominant reactive oxygen species generated upon the irradiation of porphyrins. Therefore, polystyrene was the most promising candidate for our initial investigations. **To enhance clarity of our motivation, we added further details and literature in the manuscript (page 14, line 217-221; line 226-228; page 17 line 273-277; references: 25,54-58, 64,69,70).** In the future, we aim to explore how different types of ROS affect the degradation of other common microplastics, such as PE, PP, PET, and PVC. **This is now also mentioned in the conclusion part of the manuscript (page 21, line 345-347).**

(2.8) *“The “thermally switchable buoyancy” could be not working in the natural water situation when the temperature does not satisfy the need in the surface or bottom of the water.”*

Our answer: We sincerely thank the reviewer for their valuable feedback. Regarding the temperature conditions: Since water has its highest density at 4 °C, it can be assumed that the temperature at the seabed does not fall below this value. Literature studies indicate that pNIPAM exhibits even stronger swelling at such low temperatures, which could further enhance microplastic uptake. **We added these references to the main manuscript (page 8 line 116, reference 45,46).** Concerning surface water temperatures, it is important to note that, under sufficient solar irradiation, the gel can reach temperatures of up to 50 °C despite the cooling effect of the surrounding water. This should facilitate the thermal collapse of the gel and promote microplastic degradation. However, in cloudy conditions or extremely cold environments, such as the Arctic, the applicability of the system may be limited. We acknowledge that our system still requires optimization for real-world applications. However, **the primary focus of this study was to demonstrate the autonomous ascent and descent of the gel, with microplastic degradation serving as a potential model application. To clarify the focus of our study also for potential future readers of the manuscript, we revised the manuscript accordingly study (page 2, line 15-19 and line 21-22; page 4, line 64-66; page 7, Scheme 2; page 20 line 334 to page 21 line 343)**

(2.8) *“Did author also consider the types, density, size and shape of microplastics? Can the ROS-induced decompose all microplastics? Is the time costed of microplastic decomposition same to gel flotation?”*

Our answer: We appreciate the reviewer's insightful comments. For the selection of microplastic samples, we used 2 μm polystyrene beads with a density of 1.05 g/cm^3 and a crosslinking degree of 2%, as these properties are representative of potential microplastics found at the bottom of aquatic environments. We performed dynamic light scattering experiments, showing that no small fragments, such as nanoplastics, leave the hydrogel (Supporting Information, page 24, Figure S5h). This indicates that at least particles equal or smaller than 2 μm are attacked by ROS. The photosensitizer was specifically chosen to be porphyrin-based, ensuring the production of singlet oxygen as the reactive oxygen species, which is particularly effective in degrading polystyrene. Other types of ROS may be more suitable for the degradation of different microplastics, such as non-aromatic polymers. It is worth noting that our system allows for a flexible adaptation to other ROS-generating agents, making it potentially applicable to a broader range of microplastic types. Since ROS act in a relatively non-selective manner, we do not expect the morphology of the particles to significantly influence their degradability. However, morphology is likely to impact the degradation rate, as certain structures may provide a higher surface-area-to-volume ratio, increasing the available reaction sites. This hypothesis is also supported by recent literature. **To enhance the clarity of our manuscript and incorporate the valuable feedback from the reviewer, we have revised the text accordingly and added relevant details and references (page 14, line 216-221; line 226-229; page 17 line 273-277; references: 25,54-58, 59-65).**

We appreciate the reviewer's question regarding whether the microplastic decomposition occurs within the flotation time. The flotation time is influenced by several environmental factors, including water temperature, ambient temperature, and solar irradiation. Under controlled laboratory conditions (20 °C, irradiation with a solar simulator), flotation times of approximately 40–50 minutes were achieved. However, under real environmental conditions, this duration is expected to be longer due to generally lower solar irradiation and water temperatures. For our irradiation experiments, the samples were exposed to light for 2 hours, which suggests that the flotation time falls within a reasonable range for effective microplastic degradation. Additionally, even if complete degradation does not occur within a single cycle, any remaining microplastics will undergo further decomposition in subsequent cycles, ensuring continued removal over time.

(2.8) *“Several specific comments:*

(1) Section of Main Repeated words is suspected in the sentence of “The high stability and low biodegradability.....”

(2) Scheme 1. The full name of nanoporous organosilica particles mentioned once is enough, and the abbreviation of NOPs should be used in the next. Please check the whole manuscript, and avoid the similar issues.?"

Our answer: We thank the reviewers for bringing these errors to our attention. **We therefore have revised this section in the manuscript (page 2, line 34-35).** Also, **we have corrected the usage of abbreviations in the whole manuscript.**

REVIEW 3

(3.1) *"This manuscript presents a multiresponsive hydrogel shuttle for microplastic removal from water. Notably, the hydrogel scaffold's combination of various functional units enables multiple responses for microplastic capture. This composite structure resembles a 3D micromotor capable of capturing a substantial amount of microplastics, albeit requiring additional fuel, such as a high-concentration glucose solution, to propel its motion. The hydrogel's descending motion is further facilitated by photothermal shrinkage of the temperature-responsive scaffold. Despite an intriguing concept, the complex stimuli required to activate and drive the hydrogel shuttle hinder its practical field application. Moreover, the manuscript lacks a continuous demonstration of the proposed shuttle (as depicted in Scheme 1). Furthermore, the claim of microplastic degradation through photocatalytic ROS generation is highly questionable, considering the well-established challenges in photocatalytically decomposing microplastics (e.g., Environ. Sci. Technol. 37, 2003, 4494 using TiO₂ and Cu-phthalocyanine photocatalyst). Additionally, the manuscript contains numerous grammatical errors and is written in a conversational style, impeding evaluation. Overall, this manuscript appears unsuitable for publication in Nature Communications. The following are additional comments for the authors' future submission elsewhere:"*

Our answer: Thank you for your thorough reading of our manuscript and for the constructive feedback. We appreciate the time and effort invested in the review, as it has helped us recognize that the main focus of our work was not sufficiently clear. The primary goal of our study was to utilize the extreme flexibility of our organosilica particles to develop a system capable of autonomously ascending and descending. The potential application of this system for microplastic degradation is an additional aspect rather than an immediate real-world implementation. **We have revised the manuscript accordingly to clarify the focus of our study (page 2, line 15-19 and line 21-22; page 4, line 64-66; page 7, Scheme 2; page 20 line 334 to page 21 line 343).**

Regarding the glucose concentration, we acknowledge the reviewer's concern that the D-glucose concentration in real-world systems may be lower than that used in our

study. However, we think that this does not compromise the system's autonomy but rather affects the time required for the shuttle to ascend.

Concerning the photodegradation of microplastics, we recognize the ongoing debate in the literature. The paper cited by the reviewer demonstrates that microplastics can undergo photodegradation using TiO_2 and Cu-phthalocyanine, covering nearly the entire solar spectrum. However, a noted drawback was the production of toxic byproducts due to incomplete mineralization. We see three key differences between our study and the cited work. First, in our study, the hydrogels are irradiated in dispersion, which increases material diffusion and could enhance degradation efficiency. In contrast, the cited study irradiated polystyrene films deposited on a titanium surface, where degradation products could quickly desorb and were no longer available for subsequent decomposition reactions. In our system, microplastics and their degradation products likely have a longer residence time in the BDS hydrogel, as hydrophobic organic compounds preferentially remain in the gel rather than dispersing into the surrounding water—a behaviour we demonstrated in our 2023 publication (**reference 32**). Lastly, while the cited study primarily used superoxide and hydroxyl radicals, our system predominantly generates singlet oxygen through porphyrin. **We have revised the manuscript to better reflect the current state of research and our hypothesis, and we have included the relevant literature (page 14, line 216-221; page 16, line 254 to page 18, line 294; reference 25, 54-58, 64-70). Finally, we have carefully revised the entire manuscript to improve readability and correct grammatical errors.**

- (3.2)** *“The authors claim that ROS generated by the tethered porphyrin unit can fully degrade PS microplastics. However, the degradation of rigid PS microplastics through porphyrin-based photocatalytic treatment is highly doubtful. The photothermal effect of porphyrin units may loosen the hydrogel-microplastic interaction, leading to desorption. To confirm microplastic photocatalytic decomposition, control experiments are necessary: 1) exposing a PS microplastic solution with the porphyrin unit to UV light and 2) detecting corresponding decomposition residues (e.g., CO₂ concentration in headspace, TOC in water, and microplastic mass change in the solution).”*

Our answer: We are incredibly thankful for the reviewer's constructive feedback. We sincerely appreciate the reviewer's insightful comments and acknowledge the need for a more detailed analysis of the microplastic degradation process. **In response, we have conducted additional experiments (DLS, MAS-NMR, mass loss in solution) to further support our findings** While we do not have access to a CO₂ sensor with the required accuracy, nor the means to perform TOC analysis, **we instead measured**

dynamic light scattering to confirm that no micro- or nanoplastics escaped from the BDS hydrogel (page 16, line 263-266; Supporting Information Fig. S5h,j). Additionally, we assessed changes in microplastic mass by drying the solution and weighing the residual microplastics. The measured values showed good agreement with the microplastic uptake by the BDS hydrogel, as determined by TGA. Although we cannot directly confirm the complete conversion of microplastics to CO₂, our results clearly indicate that at least partial degradation to oxygenates must have taken place, since no microplastic remains in the Gel (TGA, Raman), but also no microplastic can be detected in the washing solution (DLS). To reflect these findings more accurately, we have revised the manuscript (page 16 line 254 to page 17 line 277).

We appreciate the reviewer's concern **regarding the potential desorption of microplastics due to the photothermal effect of the porphyrin units**. However, we believe that the photothermal effect of polydopamine, given its black color, is significantly stronger than that of porphyrin, suggesting that the thermal effect of porphyrin alone can be considered negligible. **To investigate this, we conducted a reference experiment** using two identical BDS hydrogels, one of which lacked the porphyrin photosensitizer (**page 15, Figure 3f**). After irradiation, microplastics were no longer detectable in the porphyrin-containing sample, whereas they remained present in the control sample.

Both hydrogels (with and without porphyrin) were washed after microplastic uptake by repeated collapsing and swelling in water, a process carried out at 40 °C. If thermally induced desorption had a significant effect, no microplastics would have remained on either hydrogel after washing. However, since only the porphyrin-containing hydrogel exhibited a reduction in polystyrene content after subsequent irradiation, we infer that microplastic removal is driven by photocatalytic degradation rather than thermal desorption. **We have revised the manuscript accordingly (page 16, line 254 to page 17, line 277) to enhance clarity and appreciate the reviewer's insightful feedback on this point.**

- (3.3)** *"The authors claim hydrogel scaffold stability during experiments, yet the provided structural analysis is insufficient. XPS, ICP (for Pt), and SEM analyses on the cycled hydrogel, including quantitative analysis of tethered Pt, PNIPAM, and porphyrin units, should be included."*

Our answer: We thank the reviewer for the constructive feedback, and agree that the material characterization after the irradiation needed to be optimized. **Therefore, we have provided additional structural analysis to support our claim and revised the**

manuscript accordingly (page 17 line 278 to page 18 line 294; Supporting Information Figure S5k,l). Specifically, we conducted **EDX measurements**, which confirmed that platinum, silicon, and sulfur (from porphyrin and GOx) were still present in the cycled hydrogel, ensuring the stability of the scaffold. Furthermore, no optical differences were observed in the **SEM images** before and after irradiation, and **MAS solid-state NMR** analysis showed no changes in the hydrogel composition upon cycling. These results, together with the **infrared spectroscopy data (supporting information Fig. S5i)**, demonstrate the integrity of the shuttle gel and its ability to withstand repeated use in the microplastic degradation process. We believe these analyses now provide sufficient evidence for the stability of the hydrogel during the experiments and thank the reviewer for improving our manuscript.

- (3.4)** *“The mechanism by which the PDA coating prevents hydrogel scaffold degradation is unclear. The intact hydrogel structure (as claimed by the authors) might indicate microplastic detachment rather than decomposition.”*

Our answer: We appreciate the reviewer’s insightful comment regarding the mechanism by which the PDA coating prevents hydrogel scaffold degradation. To clarify, polydopamine is a well-known radical scavenger that neutralizes reactive oxygen species (ROS). This protective effect ensures that the pNIPAM chains remain intact, as ROS primarily attack the polystyrene component, which is not shielded by the polydopamin layer. Additionally, the conjugated π -system of PDA enhances π - π interactions with polystyrene, increasing microplastic adsorption onto the hydrogel surface. **To further support our conclusions and improve the manuscript’s clarity, we have included additional details and references (Page 8; line 107-112; Reference 25,42-44) as well as modifications to Figure 1a (page 9)** Regarding the possibility of thermal desorption, we conducted a control experiment using a BDS-gel without a ROS producer but otherwise identical conditions. The results demonstrate that microplastic reduction occurs exclusively in the presence of a photosensitizer, despite both gels being PDA-coated and exposed to irradiation-induced heating. This indicates that the observed microplastic decrease is due to ROS-driven degradation rather than thermal effects. **Additionally, we conducted Dynamic Light Scattering (DLS) experiments after UV irradiation, which revealed no detectable microplastic particles outside the hydrogel. To improve clarity, we added the explanation and data to the manuscript (page 16, line 254 to page 17, line 277) and the Supporting Information (page 24 Figure S5h)** We hope these clarifications address the reviewer’s concerns and appreciate their valuable feedback.

- (3.5)** *“Figure S1f: Please clarify how the hydrogel is regenerated through water washing.”*

Our answer: We sincerely appreciate the reviewer's comment, as it highlighted a lack of clarity in our manuscript. The washing with water was conducted not to regenerate the hydrogel from microplastics, but to demonstrate that the microplastic remains securely embedded within the hydrogel, even during swelling and deswelling cycles and heat treatment. **We have revised the manuscript (page 10, lines 145-149) and the Supporting Information (page 9, lines 162-164) to ensure clarity.**

(3.6) *“Figure 3: Ensure the figure description accurately matches the figure number.”*

Our answer: The reviewer was right. The figure description contained an error **that was corrected by us (page 15 line 240 to page 16 line 247)**

Point to Point answer to our Manuscript (NCOMMS-24-36108-T)

"Comments given by the reviewers"

and **our answers** (*, the page and line number given herein are regarding the marked manuscript)

REVIEW 1

(1.1) *"Define the abbreviation 'pNIPAM' in full within the abstract."*

Our answer: We thank the reviewer for the attentive reading of the manuscript and the positive feedback. **We changed the manuscript accordingly (page 2, line 19-20).**

(1.2) *"Present a plot of zeta potential versus pH for both BDS Gel and PS beads."*

Our answer: We thank the reviewer for their valuable suggestion. As recommended, **we measured the zeta** potential as a function of pH for the PS bead dispersions; **the results are now included in the Supplementary Information (Figure S1e)** and align well with literature values (−10 mV at pH 3 to −33 mV at pH 9). Direct measurements for the BDS gel were not feasible due to its non-dispersible nature, but literature values (0 mV at pH 3 to −25 mV at pH 9) for polydopamine-based surfaces **are cited in the revised manuscript (page 8, lines 116–120; refs. 45–47)**. Despite negative zeta potentials for both materials, adsorption is still effective due to salt-induced screening and non-electrostatic interactions such as π – π stacking and hydrophobic effects.

(1.3) *"Please provide the SEM image of the BDS hydrogel after MP uptake, maintaining the same scale as the pre-uptake images."*

Our answer: We thank the reviewer for highlighting this important detail. **We therefore corrected this error in the manuscript (page 9, Figure 1).**

(1.4) *"Provide an estimate of the operational costs associated with MP removal using the BDS hydrogel."*

Our answer: We thank the reviewer for the valuable comment. In response to the request for an estimate of the operational costs associated with microplastic removal using the BDS-gel, we provide the following information:

Material Costs: The estimated cost for producing **1 gram of BDS-gel** is approximately **318 €** without accounting for energy consumption. When energy costs are included, the total material cost increases to **588 € per gram**. It is important to note that these costs reflect a small-scale academic synthesis, where the costs of labor, equipment, and scale are higher. For industrial-scale production, these material costs could be significantly reduced due to economies of scale, improved manufacturing processes, and optimized resource utilization.

Operational Costs: Since the BDS hydrogel system operates autonomously, the ongoing operational costs are minimal. The primary operational expenses involve transporting the gel to the required location and occasionally checking its functionality. The system relies on natural resources, such as sunlight and glucose, which are freely available and do not incur additional costs. Therefore, the operational costs are expected to be very low compared to the material costs. **We have added the cost calculation to the supporting information (Fig. S6c) as well as the manuscript (page 20, line 352-354).**

(1.5) *“Clarify the atomic weight percentage of Pt, considering its minimal intensity in Fig. S5(k).”*

Our answer: We thank the reviewer for this helpful comment. **We added the atomic and weight percentages to the supplementary information (Figure S5k)**

(1.6) *“Rectify typographical errors found in the figure captions of the Supplementary Information.”*

Our answer: We thank the reviewer for the careful review. **We have rectified the typographical errors found in the figure captions of the Supplementary Information.**

(1.7) *“It would be good if the author could add the scale bar inside the TEM and SEM images instead of writing as a figure caption.”*

Our answer: We agree with the reviewers' recommendation and integrated the scale bar inside the TEM and SEM micrographs (Figure 1-3, Figure S2a,b and Figure S5e)

REVIEW 2

(2.1) *“This paper has been carefully addressed according to reviewers' comments, and current version is suitable for publication.”*

Our answer: We thank the reviewer for the very positive feedback on our manuscript. Their comments were invaluable in helping us improve the paper, and we are pleased to hear that the current version is suitable for publication. We greatly appreciate the time and effort dedicated to reviewing our work.

REVIEW 3

“While the authors have addressed some of the previous concerns, the manuscript, in its current state, does not demonstrate sufficient novelty for publication in Nat. Commun. The concept of fuel-driven micromotors is well established, and the proposed innovation, a descending motion

achieved through photothermal shrinkage of a temperature-responsive scaffold, lacks robust experimental validation.

Our answer: We thank the reviewer for the continued evaluation of our manuscript and appreciate the opportunity to further clarify the core innovation our work. **While we acknowledge that fuel-driven micromotors are well established in the literature, our study introduces a conceptually different approach:** a soft, fuel-driven system capable of cyclic vertical motion—both ascent and descent—enabled by internal material design alone, without external magnetic or electric guidance. The key innovation lies in the reversible gas exchange within nanoporous organosilica particles, which, when coupled with a temperature-responsive hydrogel scaffold and embedded photosensitizer, enables a unique buoyancy control mechanism. Although individual components have been previously studied, **to our knowledge this level of functional integration—combining autonomous locomotion, selective capture, and spatially localized degradation—has not been previously reported.** (see page 4, lines 70–73 and page 20, line 356 to page 21, line 375).

While we recognize that novelty assessments can be subject to interpretation, we respectfully maintain that the degree of functional integration and autonomy demonstrated in our system has not, to the best of our knowledge, been reported in the literature. If the reviewer is aware of prior work that demonstrates a comparable combination of design and functionality, we would be grateful for such references. **We have further revised the manuscript to better articulate this distinction** and hope that these clarifications help to convey the contribution more effectively.

- (3.1) *Specifically, the authors fail to provide compelling evidence of continuous shuttle operation, as depicted in Scheme 1, casting doubt on the practical viability of their concept. Furthermore, the claim of PS photodegradation by the reported materials remains highly questionable. While presented as a proof-of-concept, the presented data is insufficient to support this assertion. The new TGA and DLS analyses are inadequate for characterizing the comprehensive decomposition of PS. Detecting the degradation of 15 mg of PS within a 150 μ L, 10 wt% solution using these techniques is inherently challenging. Moreover, typical DLS analyses of microplastic degradation demonstrate surface weathering and gradual size reduction during photocatalytic treatment (J. Environ. Chem. Eng. 2022, 10, 108195)."*

Our answer: We thank the reviewer for his valuable feedback. We fully agree that TGA has limitations when it comes to detecting subtle chemical changes or partial surface degradation. However, we would like to clarify that, under the specific conditions of our experiment, TGA provides sufficient sensitivity to detect any significant residual mass of undegraded polymer. The instrument used offers high mass resolution and has proven reliable for detecting such differences in similar contexts. As the reviewer correctly notes, detailed DLS studies on PS degradation typically observe surface weathering and gradual size reduction over time (as described in J. Environ. Chem. Eng. 2022, 10, 108195). In our case, DLS was employed not to

characterize the degradation pathway or fragmentation profile in detail, but rather to confirm the absence of PS micro- or nanoparticles in the supernatant following irradiation. Thus, the intent was not to track morphological evolution, but to provide a qualitative indication that no detectable PS fragments are released into the surrounding solution. We believe that, for this specific purpose, DLS remains an appropriate tool. **To acknowledge the limitations and potential pitfalls of the method, we have cited the reference kindly provided by the reviewer and discussed these considerations explicitly in the revised manuscript (page 16, line 271 to page 17, line 278; reference 72).**

- (3.2) *“The current study’s photodegradation experiments, conducted over a mere two hours under 420 nm irradiation, are insufficient to achieve significant PS degradation, despite the authors’ claims regarding singlet oxygen generation. The inherent difficulty of microplastic degradation in aqueous environments is well-documented, as evidenced by numerous studies, including Environ. Sci. Technol. 37, 2003, ACS Sustain. Chem. Eng. 2023, 11, 10688, ChemSusChem 2024, 17, e202301350, Solar RRL 2023, 7, 2300411, and many more. To strengthen their claims, the authors should conduct degradation tests using standardized, dyed PS microplastic dispersions, which would allow for direct visual confirmation of microplastic fate. Additionally, incorporating photocatalytic degradation experiments using model organic pollutants (e.g., dyes) would provide valuable insights into the efficiency of the developed materials.”*

Our answer: We sincerely thank the reviewer for their thoughtful and insightful comments, as well as for highlighting key references concerning the photodegradation of polystyrene in aqueous environments. We deeply appreciate the opportunity to reflect on these important studies and your valuable suggestions, which have greatly enhanced the clarity and depth of our manuscript. We see that our decision is supported that a mechanistic investigation of PS decomposition by ROS is not the focus in the current paper. Still, in line with the recommendations given by the reviewer, it could be a topic for a follow-up project.

After a thorough review of the relevant literature and careful consideration of your observations, we acknowledge that a two-hour exposure to 420 nm irradiation may appear unexpectedly short for achieving substantial polystyrene (PS) degradation. We hypothesize that the enhanced degradation efficiency observed in our system may be attributed to a unique advantage of the BDS gel — its intrinsic porosity. In contrast to many previously reported systems, such as microswimmers or TiO₂ particles, which are typically non-porous, the BDS gels possess hierarchical porosity across multiple length scales (nm-scale from the NOPs and μm-scale from the hydrogel). **This multiscale porosity likely facilitates the prolonged retention of microplastics within the gel matrix and in close proximity to active degradation sites, thereby potentially increasing their exposure time to reactive species and enhancing photodegradation.** Upon desorption, these microplastics are less likely to escape into the surrounding solution and may instead re-adsorb at new locations within the gel, where they can continue to degrade. Moreover, it is plausible that degradation byproducts are retained within the hydrogel network for extended durations, further contributing to the observed efficiency.

These synergistic effects may collectively explain the material's superior performance in this context. **In-line with the reviewer's comment, we have now included the references kindly suggested by the reviewer (references 68, 72, 75–77) and have revised the manuscript accordingly (page 17, lines 287–297) to incorporate these hypotheses, along with additional supporting literature (references 78-81).**

We also greatly appreciate the reviewer's excellent suggestion to employ standardized, dyed PS microplastics or fluorescent dyes as model pollutants. This is indeed a highly valuable approach and holds great promise for future studies. However, we respectfully note that such an investigation would extend beyond the current scope of our manuscript. The primary aim of this work is to demonstrate the BDS gel's autonomous, buoyancy-driven vertical motion, which serves as the core innovation of the system. The observed microplastic degradation is presented here as an initial application—a proof of concept—that highlights one potential use of the material. Nonetheless, we fully agree that a more comprehensive investigation using model pollutants would be an important direction for future work.

(3.3) *“Compounding these scientific concerns is the persistent lack of improvement in the manuscript's clarity and language, including the presence of typographical errors within highlighted revised sections. This further detracts from the overall quality of the work.”*

Our answer: We thank the reviewer for the attentive reading of the manuscript and the valuable feedback. We sincerely apologize for the persistent issues related to the manuscript's clarity and language, as well as for the typographical errors in the revised sections. We fully recognize that these issues detract from the overall quality of the work and are committed to addressing them thoroughly. In response to your comment, **we have carefully reviewed the manuscript to correct any typographical errors and improve the clarity of the language throughout.** We believe these revisions significantly enhance the readability and scientific communication of the paper, and we hope that the updated version will meet your expectations.